# Diversity of Epithelial-Mesenchymal Phenotypes in Circulating Tumour Cells from Prostate Cancer Patient-Derived Xenograft Models

**DOI:** 10.3390/cancers13112750

**Published:** 2021-06-01

**Authors:** Sara Hassan, Tony Blick, Erik W. Thompson, Elizabeth D. Williams

**Affiliations:** 1Faculty of Health and Institute of Health & Biomedical Innovation (IHBI), School of Biomedical Sciences, Queensland University of Technology (QUT), Brisbane City, QLD 4000, Australia; sara.hassan@hdr.qut.edu.au (S.H.); tony.blick@qut.edu.au (T.B.); 2Translational Research Institute (TRI), Brisbane, QLD 4102, Australia; 3Australian Prostate Cancer Research Centre—Queensland (APCRC-Q), Brisbane, QLD 4102, Australia; 4Queensland Bladder Cancer Initiative (QBCI), Brisbane, QLD 4102, Australia

**Keywords:** circulating tumour cell (CTC), epithelial-mesenchymal plasticity (EMP), metastasis, prostate cancer, patient-derived xenograft (PDX), kallikrein-related peptidase 3 (*KLK3*), prostate-specific antigen (PSA), *SERPINE1*, plasminogen activator inhibitor-1 (*PAI-1*)

## Abstract

**Simple Summary:**

Spread of prostate cancer to other parts of the body is responsible for the majority of deaths. Tumour cell epithelial mesenchymal plasticity (EMP) increases their metastatic potential and facilitates their survival in the blood as circulating tumour cells (CTCs). The aim of this study was to molecularly characterise CTCs in a panel of prostate cancer patient-derived xenografts using genes associated with epithelial and mesenchymal phenotypes, and to compare the EMP status of CTCs with their matched primary tumours. The study highlights high heterogeneity in CTC enumeration and EMP gene expression between tumour-bearing mice and within individual blood samples, and therefore caution should be taken when interpreting pooled CTC analyses. Critically, tumour cells were present in the epithelial-mesenchymal hybrid state in the circulation. The study also demonstrates that there is high variation in CTC size, which would introduce sample bias to size-based CTC isolation techniques.

**Abstract:**

Metastasis is the leading cause of cancer-related deaths worldwide. The epithelial-mesenchymal plasticity (EMP) status of primary tumours has relevance to metastatic potential and therapy resistance. Circulating tumour cells (CTCs) provide a window into the metastatic process, and molecular characterisation of CTCs in comparison to their primary tumours could lead to a better understanding of the mechanisms involved in the metastatic cascade. In this study, paired blood and tumour samples were collected from four prostate cancer patient-derived xenograft (PDX) models (BM18, LuCaP70, LuCaP96, LuCaP105) and assessed using an EMP-focused, 42 gene human-specific, nested quantitative RT-PCR assay. CTC burden varied amongst the various xenograft models with LuCaP96 having the highest number of CTCs per mouse (mean: 704; median: 31) followed by BM18 (mean: 101; median: 21), LuCaP70 (mean: 73; median: 16) and LuCaP105 (mean: 57; median: 6). A significant relationship was observed between tumour size and CTC number (*p* = 0.0058). Decreased levels of kallikrein-related peptidase 3 (*KLK3*) mRNA (which encodes prostate-specific antigen; PSA) were observed in CTC samples from all four models compared to their primary tumours. Both epithelial- and mesenchymal-associated genes were commonly expressed at higher levels in CTCs compared to the bulk primary tumour, although some common EMT-associated genes (*CDH1*, *VIM*, *EGFR*, *EPCAM*) remained unchanged. Immunofluorescence co-staining for pan-cytokeratin (KRT) and vimentin (VIM) indicated variable proportions of CTCs across the full EMP axis, even in the same model. EMP hybrids predominated in the BM18 and LuCaP96 models, but were not detected in the LuCaP105 model, and variable numbers of KRT^+^ and human VIM^+^ cells were observed in each model. *SERPINE1*, which encodes plasminogen activator inhibitor-1 (PAI-1), was enriched at the RNA level in CTCs compared to primary tumours and was the most commonly expressed mesenchymal gene in the CTCs. Co-staining for SERPINE1 and KRT revealed SERPINE1^+^ cells in 7/11 samples, six of which had SERPINE^+^KRT^+^ CTCs. Cell size variation was observed in CTCs. The majority of samples (8/11) contained larger CTCs ranging from 15.3 to 37.8 µm, whilst smaller cells (10.7 ± 4.1 µm, similar in size to peripheral blood mononuclear cells (PBMCs)) were identified in 6 of 11 samples. CTC clusters were also identified in 9/11 samples, containing 2–100 CTCs per cluster. Where CTC heterogeneity was observed in the clusters, epithelial-like cells (KRT^+^VIM^−^) were located on the periphery of the cluster, forming a layer around hybrid (KRT^+^VIM^+^) or mesenchymal-like (KRT^−^VIM^+^) cells. The CTC heterogeneity observed in these models emphasises the complexity in CTC isolation and classification and supports the increasingly recognised importance of the epithelial-mesenchymal hybrid state in cancer progression and metastasis.

## 1. Introduction

Prostate cancer is the second most commonly diagnosed cancer in men worldwide [1]. Despite best clinical practice, such as treatment with curative intent or active surveillance, many patients eventually succumb to metastasis. Although men with localised prostate cancer have high 5-year survival and low mortality rates, men with metastatic disease have very high mortality rates. Median survival is 6.6 years from diagnosis of non-castrate metastases, and significantly shorter (1–3 years) when in the castrate-resistant state [2,3]. Tumour cells can dissociate from the primary tumour and enter the blood stream to become circulating tumour cells (CTCs). Some of these cells survive in the circulation and exit the blood stream at a distal site, and a proportion of these cells colonise new locations to form metastatic deposits. The role of CTCs in metastasis and their association with clinical disease progression and recurrence has been well established in prostate cancer [4,5,6], and they have also been detected in many other types of solid cancers [7,8,9]. While the majority of studies have focused on CTC enumeration, molecular characterisation of CTCs may enhance the clinical utility of CTCs and elucidate the molecular mechanisms leading to tumour cell dissemination, CTC survival under unfavourable conditions, and formation of metastases. These analyses thus have potential prognostic, predictive and therapeutic relevance.

Prostate-specific antigen/kallikrein-related peptidase 3 (PSA/KLK3) is a serine protease produced by epithelial cells in the prostate gland that is used as a clinical serum biomarker to detect and monitor prostate cancer [10]. Elevated serum PSA levels are a trigger for investigation of possible prostate cancer. However, despite being a very sensitive biomarker, detectable serum PSA is not specific for prostate cancer and further evidence is required for a definitive diagnosis. Tumour biopsy is the gold standard for the diagnosis of prostate cancer; however, repeated tumour biopsies for monitoring treatment progress are not clinically acceptable and serum PSA and imaging are used instead. New ways to monitor prostate cancer progression are required to further optimise disease management. It has been observed that CTC detection can precede increases in blood PSA levels in men during recurrence of prostate cancer, thus monitoring CTCs could potentially lead to earlier implementation of effective intervention(s) to improve treatment outcome and overall survival [6,11]. Furthermore, not all prostate cancer is PSA-positive or readily detectable on imaging. There is also a need for better characterisation of the disease using alternative approaches. There is now an array of treatment options for men with prostate cancer. Precision medicine approaches that help identify the right treatment for the right man at the right time play a key role in optimising clinical planning and minimising treatment failures and unwanted side-effects. CTC monitoring provides such an opportunity [12,13,14].

Currently, the CellSearch^®^ system is the only test that is approved for detection of CTCs in prostate cancer by the United States Food and Drug Administration [15]. This test relies on the expression of epithelial cell markers EPCAM and epithelial cytokeratins (8, 18, and/or 19) by prostate cancer cells. However, CTC phenotype may be altered via epithelial mesenchymal plasticity (EMP), leading to loss of epithelial characteristics and/or gain of mesenchymal features [16,17], which may impact on CTC detection using this technology.

EMP is proposed to play a vital role in the metastatic cascade. Cancer cells can undergo epithelial mesenchymal transition (EMT), which results in a loss of cell adhesion and cell polarity, and causes enhanced translocation capacity [18,19]. This enables cancer cells to detach from the primary tumour and intravasate into the blood circulatory system. These CTCs have the ability to undergo the reverse process as well, via mesenchymal epithelial transition (MET), through which they become more epithelial in nature and which in turn may enhance proliferation at the newly established site. However, it is now well established that CTCs do not always undergo complete transition, and thus may exist in a hybrid state whereby cells express both epithelial and mesenchymal markers [20,21]. This hybrid state has been shown to be more metastatic than epithelial or mesenchymal states [20,22,23]. EMP may thus promote the metastatic cascade by enabling the cells to adjust to changing microenvironments and enhance cell survival in the circulation and at distant sites [24].

Pre-clinical patient-derived xenograft (PDX) models are used to understand molecular mechanisms and test potential cancer therapies [25]. They are generated by implanting clinical cancer biospecimens (most commonly obtained from biopsy or surgical specimens) in immuno-compromised hosts (typically mice), and may sometimes by subsequently serially passaged by transferring a small piece of the resultant tumour to a new host. When compared to cancer cell lines and cancer cell line xenograft models, PDX models are more genetically stable and are able to recapitulate the characteristics of their donor across numerous passages [26,27]. Such model systems are vital in translational cancer research for understanding cancer development and progression in the clinical setting. PDX models also allow the study of CTCs, disseminated tumour cells (DTCs) and circulating tumour DNA (ctDNA) in blood drawn from these mice.

We have previously employed a human-specific, tandem nested RT-qPCR-based approach to quantify RNA expression of a panel of 40 genes in xenograft tumours and their paired CTCs and DTCs in a PDX and a breast cancer cell line xenograft model [28]. We observed altered expression of both epithelial and mesenchymal-associated genes, consistent with an EMP hybrid state of CTCs. While breast cancer CTCs and DTCs have been studied in several different PDX models [29,30,31,32], to our knowledge, no such study has been performed using prostate cancer PDXs.

The current study aims to expand our understanding of prostate cancer by studying PDX-derived CTCs using both molecular PCR-based and cell staining approaches, with a particular focus on EMP characteristics that may regulate tumour cell dissemination.

## 2. Results

### 2.1. Identification of CTCs Using Immunofluorescence Analysis

Bloods from representative samples of mice across all 4 xenografts (11 total) were assessed for the presence of human vimentin (VIM) and/or cytokeratin 8/18/19 (KRT) -positive cells (Figure 1 and Appendix A). Use of the V9 human-specific VIM antibody enabled identification of human prostate cancer cells and not other murine mesenchymal cells that would also be VIM^+^. It also provided context for the morphology and size of the KRT^+^ cells. One negative control blood sample (from a mouse that did not grow a PDX tumour) was also included in the analysis. Only small, KRT^−^VIM^−^ DAPI-stained nuclei were observed in this negative control sample (Appendix A). All PDX-bearing mouse blood samples had at least one VIM^+^ and/or KRT^+^ cell. The total number of CTCs per mL of blood varied from 1 to 1294, with a median of 7 CTCs per mL (Table 1). Characteristics of the CTC populations varied within PDX models, with samples presenting with only VIM^+^ cells, only KRT^+^ cells, or a mixture of VIM^+^ and KRT^+^ cells. Hybrid cells (VIM^+^KRT^+^) were also observed in 6 of 11 samples with numbers of hybrid CTCs ranging from 1 to 89 per mL of blood. Hybrid cells were most often detected in the BM18 and LuCaP96 models, and never detected in the LuCaP105 model. Most CTCs were almost twice the size of normal peripheral blood mononuclear cells (PBMCs, 10–15 µm [33]), ranging from 15.3 to 37.8 µm in diameter and had larger nuclei. However, most samples (2/3 BM18, 2/3 LuCaP70, and both LuCaP105 samples) also contained CTCs that had the same diameter as PBMCs (10.7 ± 4.1 µm compared to 10–15 µm, respectively), although with a higher nuclear size to cytoplasmic ratio. No differences were seen in the range of CTC numbers between models. CTC clusters were detected in four samples (BM18, 2x LuCaP70 and LuCaP96) and contained VIM^+^, KRT^+^ or VIM^+^KRT^+^ cells; these cells were always large. In clusters containing a mix of these cell phenotypes, KRT^+^ cells were always located on the periphery while VIM^+^ cells were surrounded by VIM^+^KRT^+^ hybrid or KRT^+^ cells. All tumour cells in these clusters had large nuclei. The number of tumour cells per cluster varied from 2 to 100 CTCs. No CTC clusters were observed in the LuCaP105 model.

### 2.2. RNA-Based Pre-Screening of Blood Samples for the Presence of CTCs

Bloods and paired tumours were progressively accumulated during routine passaging of each PDX, and gene expression profiles were assessed in batches. A pre-screen on all the blood samples was performed using human-specific tandem-nested RT-qPCR for housekeeper gene (HKG) *RPL32* and *KLK3*, an abundant, human-specific prostate epithelial cell transcript, to screen for the presence of CTCs prior to employing the full assay panel (Figure 2 and Appendix A). Twenty samples (20/71) did not have any detectable *RPL32* or *KLK3* transcripts and were not included in further analyses. Twenty-nine percent of *RPL32*-positive samples did not show detectable *KLK3* (15/51), whereas only one *KLK3*-positive sample did not have detectable *RPL32* expression. Based on our previous study in breast cancer mouse models [28], we aimed to include 10–15 samples per PDX model in our analysis. In the current study, mouse samples were collected sequentially and a pre-screen was performed for the presence of CTCs. All samples that were positive for *RPL32* were analysed using a 42-gene panel (BM18, *n* = 14; LuCaP70, *n* = 12; LuCaP96, *n* = 11; LuCaP105, *n* = 14).

Unlike *KLK3*, *RPL32* is also expressed in mice, so a test was performed to confirm there was no detection of mouse cells with these primers and amplification conditions using mouse mammary cell line 4T1 and human breast cancer cell line MDA-MB-231. The mouse cell line showed no amplification, whereas a significant signal was observed for *RPL32* in the human cell line (Appendix A).

### 2.3. Human-Specific RT-qPCR Profiling of Paired PDX Tumour and Blood Samples

We employed human-specific RT-qPCR profiling to further characterise the phenotypes of CTCs and their respective paired PDX tumour. A complete 42-gene tandem-nested RT-qPCR assay was run on all *RPL32*-positive blood and matched tumour pieces. In this panel, *SNAI1*, *VIM*, *NOTCH1*, *EGFR*, *FN1*, *SERPINE1*, *SNAI2*, *VCL*, *IGF1R*, *RRAS*, *FOSL1*, *MSN*, *NRP1*, *LAMC2*, *TNC*, *EMP3* and *INHBA* are mesenchymal-associated genes and *JUP*, *KRT20*, *KLK3*, *CDH1*, *GRHL2*, *EPCAM*, *BMP7*, *CLDN3*, *CLDN4*, and *CLDN7* are epithelial-associated genes [28]. A small number of hypoxia-associated genes (*APLN*, *HIF1A*, and *BNIP3*), cancer stem cell (CSC) markers (*CD24* and *CD44*), hormonal regulation (HR) genes (*ESR1*, *PGR*, and *TFF1*), other selected genes (*PPARGC1A*, *ILK*), and HKGs (*RPL32*, *GUSB*, *TBP*, *OAZ1* and *NONO*) were also included in the panel.

The number of detectable genes was compared with the signal strength for *RPL32* in each sample (Figure 3). A negative correlation was observed between *RPL32* Ct and the number of genes detected (r = −0.68), thus the higher the *RPL32* expression level, the greater the number of genes detected. Only samples with at least five detectable genes were used in further analysis (Table 2, Figure 3). Using this criterion, mice bearing LuCaP96 PDX had the highest frequency of CTC-positive samples (77%), followed by those bearing LuCaP70 (69%), LuCaP105 (60%) and BM18 (50%) tumours.

Since the number of CTCs shed into the blood from individual tumours may vary quite significantly, the gene expression observed in the blood will be affected by the CTC number present in each sample. CTC number may be influenced by a variety of factors, including tumour size and tumour cell phenotype. CTC burden at endpoint (Appendix A) was associated with endpoint tumour size (Figure 4). The estimated number of CTCs in an individual mouse ranged from as few as 1 CTC to almost 4800 CTCs, with median estimates for individual models of 21 (BM18), 16 (LuCaP70), 31 (LuCaP96) and 6 (LuCaP105) CTCs. The estimated mean CTC numbers (standard deviation shown in brackets) for each model were: BM18: 101 (120), LuCaP70: 73, (140), LuCaP96: 704 (1547), and LuCaP105: 57 (105).

Hierarchical unsupervised clustering of all the tumour and blood samples together, irrespective of PDX model, identified several major subgroups within the blood samples (Figure 5). A small number of blood samples (*n* = 6; comprising 1 or 2 samples of each PDX type) clustered directly with the tumours and showed relatively high expression of genes associated with epithelial rather than mesenchymal phenotype. Another group of blood samples (*n* = 9) clustered together and had high expression of both epithelial and mesenchymal genes consistent with a hybrid subset of CTCs in the blood. These samples also had a consistently higher number of CTCs compared to the other subgroups (Figure 5, boxed samples). Interestingly, this group was enriched for BM18 PDX-derived samples (5/11 BM18 PDXs were in this cluster) and devoid of samples from mice bearing LuCaP96 tumours. The remainder of blood samples did not show a strong epithelial or mesenchymal state using this gene panel, nor were they similar to their matched primary tumours. Tumour samples displayed a predominately epithelial prolife, with relatively high levels of expression of epithelial genes and relatively low expression of mesenchymal genes.

Scrutiny of the expression profiles of tumours and CTCs revealed several important observations (Figure 6a–d and summarised in Table 3). Expression levels were only compared in mice in which the gene could be detected, because of the limitation of detection threshold. As seen in Appendix A, with the exception of *KLK3*, *PPARGC1A* and *ESR1*, the *RPL32* RNA level (indicated as raw Ct value) was significantly lower in mice in which each gene was not detected, consistent with the likelihood that there were insufficient CTCs for an accurate measurement in most cases when the genes were not detectable.

Coincident upregulation of both epithelial and mesenchymal genes was observed in the majority of blood-derived CTC samples compared to primary tumour, while a few epithelial genes such as *CLDN7*, *CD24* and *CDH1* were downregulated in the CTC samples. Very few significant changes were noted in the LuCaP96 model, despite similar CTC burdens to the other PDX models (Figure 5). Mesenchymal-associated genes were most prominently and consistently upregulated across the other three models.


***Mesenchymal and Epithelial genes:***


*SNAI1*, *SERPINE1*, *MSN*, *NRP1* and *LAMC2* were significantly upregulated in CTCs compared to primary tumours in three of the four models, with the fourth model (LuCaP96) also showing a similar trend for *SNAI1. RRAS* and *TNC* was higher in expression in CTCs in the BM18 and LuCaP105 models, with a similar trend observed in the other two models for *RRAS. INHBA* was detected more frequently in the blood samples (7/11 BM18, 5/11 LuCaP70, 4/10 LuCaP96, 7/10 LuCaP105) than in the tumours (3/11 BM18, 3/11 LuCaP70, 2/10 LuCaP96 and 2/10 LuCaP105 tumour specimens), and accordingly, the level of expression of *INHBA* in tumours was lower than in the blood, except in LuCaP96. This only reached significance in the BM18 model. *SNAI2* was the least frequently detected gene in blood samples (1/11 BM18, 1/11 LuCaP70, 0/10 LuCaP96 and 1/10 LuCaP105), although it was detected in the majority of tumour specimens (10/11 BM18, 10/11 LuCaP70, 4/10 LuCaP96, 10/10 LuCaP105). Where detected, its expression was similar to that observed in the tumours. Interestingly, no change in expression levels of mesenchymal-associated genes *VIM*, *EGFR*, *FOSL1* and *FN1* was observed between CTCs and their primary tumours in any of the models. No significant difference in expression of *FOSL1* was observed and it was very rarely detected in any of the models (tumours: 0/11 BM18, 1/11 LuCaP70, 1/10 LuCaP96, 2/10 LuCaP105; blood: 2/11 BM18, 3/11 LuCaP70, 3/10 LuCaP96, 3/11 LuCaP105).

In the context of epithelial-associated differentiation, *KLK3* was found to be downregulated in the CTCs compared to primary tumours in all four models, and *CD24* was downregulated in BM18 and LuCaP105. *KLK3* was one of only three genes for which the CTC burden (estimated using raw Ct value for *RPL32*) was not significantly lower when it was not detected, indicating that *KLK3* was subject to strong repression. *EPCAM*, a commonly employed epithelial-associated gene in the CTC field, had CTC expression levels similar to primary tumours. *CDH1* and *CLDN7* were downregulated in the CTCs of the LuCaP105 model only. *CLDN4* was upregulated in three of the four models, with LuCaP105 being the only model lacking this change. *GRHL2* was upregulated in BM18 and LuCaP70. Other changes in epithelial gene expression that were model-specific include upregulation of *EMP3* in LuCaP70, *JUP* in LuCaP105, and *BMP7*, *KRT20* and *CLDN3* in BM18.

Thus, a dysregulated EMP was observed across the PDX models, with an increase in many mesenchymal-associated genes alongside an increase in some epithelial-associated genes. The most commonly studied genes in terms of EMP, *EPCAM*, *VIM* and *KRT20*, showed little or no change in expression in any of the models, emphasising the need to study a larger panel of genes when describing EMT phenotypes.


***Cancer stem cell markers:***


*CD24/CD44* analysis showed coordinated downregulation of *CD24* and upregulation of *CD44* in CTCs in the BM18 and LuCaP105 PDX models in comparison with their tumours (Appendix A). Upregulation of *CD44* only was observed in LuCaP70 CTCs, while there was no change in either marker in LuCaP96 CTCs.


***Anoikis, Hypoxia, Metabolism:***


Noteworthy observations of some other genes of interest for CTCs included the anoikis-suppressing marker *ILK*, which was significantly upregulated in CTCs derived from BM18, LuCaP70 and LuCaP105 PDXs, and upregulation of the metabolism-associated *PPARGC1A* in BM18 and LuCaP105 PDX CTCs. Interestingly, even though this observation was statistically significant, only two blood samples in the LuCaP105 model had any detectable expression of the *PPARGC1A* gene, meaning it was either below the detection threshold (9/11 samples) or very highly expressed (2/11 samples). *APLN*, *HIF1A* and *BNIP3* are hypoxia-associated genes; no difference in their expression was observed for *BNIP3*, but upregulation of *APLN* was seen in LuCaP70 and LuCaP105, and upregulation of *HIF1A* in BM18 and LuCaP70 CTCs.

Overall, the BM18 model had the most significant differences between gene expression in the tumours and CTCs. There was higher expression of 11 mesenchymal- and 5 epithelial-associated genes, and lower expression of one epithelial-associated gene (*KLK3*) in the CTCs compared to tumours, suggesting a hybrid-like phenotype. Similarly, LuCaP70 CTCs show higher expression of six mesenchymal-associated genes and two epithelial-associated genes, and lower expression of *KLK3*. LuCaP105 CTCs showed higher expression of seven mesenchymal- and one epithelial-associated gene, as well as expression of three epithelial-associated genes (*CDH1*, *CLDN7* and *KLK3*). The lowest number of differences was observed in LuCaP96, where the CTCs seemed quite epithelial in nature with a gene expression pattern that was similar to their primary tumours. Only one epithelial-associated gene (*CLDN4*) was more highly expressed in CTCs compared to tumours, and one epithelial-associated gene (*KLK3*) was expressed at lower levels.

Gene expression for the primary tumours across the four PDX models was similar; however, a few model-specific differences were noted (Appendix A). Mesenchymal-associated genes *EGFR* and *RRAS*, and an anoikis-associated gene (*ILK*) had high expression in LuCaP70 and LuCaP105. *VIM* was uniformly high in all LuCaP105 samples. *FN1* had high expression in only LuCaP96, while *VCL* had high expression in all models except BM18. By contrast, epithelial genes *JUP*, *CDH1*, *GRHL2*, *EPCAM*, *CLDN3*, *CLDN4*, *CLDN7* and *KLK3* were uniformly highly expressed in all models. Despite being rich in epithelial markers, the tumours appeared to have a hybrid-like phenotype due to upregulation of few mesenchymal genes. In terms of the CSC markers, *CD24* was low in LuCaP70, and *CD44* was low in BM18. The housekeeper genes were consistently highly expressed in all the models.

### 2.4. Immunocytochemical Assessment of Specific Markers

We selected plasminogen activator inhibitor-1/serine protease inhibitor serpin-E1 (SERPINE1) for further protein-level analysis in conjunction with VIM, pan-keratin 8/18/19 (KRT) and the clinically relevant kallikrein-related peptidase 3/prostate-specific antigen (PSA). *SERPINE1* expression was significantly higher in CTCs in comparison with the tumours in three of the four PDX models, with a similar trend in the fourth one. While some other mesenchymal genes also followed this pattern (*SNAI1*, *MSN*, *NRP1* and *LAMC2*), *SERPINE1* was the most commonly expressed mesenchymal gene in the CTCs in our study. To examine the context of *SERPINE1* expression, PDX blood samples were also stained for KRT (Figure 7, Table 4). Cells co-staining for SERPINE1 and KRT were observed in 6/11 blood samples and made up the majority of the CTC population in 4 of the 6 positive samples, with numbers ranging from 4 to 944 cells per mL of blood. Of these samples, one contained only co-stained cells, three also contained SERPINE1^−^KRT^+^ cells, and two comprised all three phenotypes (SERPINE^+^KRT^+^, SERPINE1^−^KRT^+^, SERPINE1^+^KRT^−^). Of the remaining five samples (negative for double staining population), one sample contained SERPINE1^−^KRT^+^ and SERPINE^+^KRT^−^ cells, one contained only SERPINE1^−^KRT^+^, and three samples did not contain any cells staining positively for either SERPINE1 or KRT. The total number of CTCs detected using SERPINE1 and KRT staining was significantly correlated with that seen using VIM and KRT staining (*p* = 0.005, r = 0.8). There was high variation in KRT^+^ and/or SERPINE1^+^-stained cell number across and within PDX models ranging from 0 to 950 CTCs per mL of blood (mean = 249, median = 26). No CTCs were detected in the LuCaP105 model samples, in which CTC numbers observed previously using VIM and KRT staining were also low (Table 4), nor in one of the LuCaP70 samples. Using this staining approach, CTC clusters were most often detected in LuCaP70 and LuCaP96, with CTC numbers ranging from 2 to 100 CTCs per cluster.

Each sample was also stained for PSA alongside either VIM or SERPINE1, based on which of the two were found to be positive in CTCs in our previous analysis, and given the limitation of only having three cytospin samples available for each blood sample (Figure 8, Table 5). CTCs (positive for PSA and/or VIM or SERPINE1) were detected in all samples with numbers ranging from 3 to 1166 CTCs per mL of blood (median = 17, mean = 181). Cells simultaneously staining for PSA and VIM/SERPINE1 were found in all samples, and made up a large portion of the total CTC population, with the number of CTCs ranging from 1 to 558 (median = 9) per mL of blood. Samples with high total CTC numbers mostly also had high numbers of hybrid cells (*p* = 0.001, r = 0.887). Cells staining for only PSA were found in 5/8 SERPINE1/PSA-stained samples, with CTC numbers ranging from 1 to 568 per mL of blood. CTC clusters were observed in 9/11 samples, including single LuCaP105 CTC clusters comprised of 2–3 cells and only expressing the mesenchymal-associated genes SERPINE1 or VIM.

## 3. Discussion

Metastatic prostate cancer remains an incurable disease. PSA is routinely used to monitor cancer progression but often fails to predict relapse if tumour burden is low or the disease evolves to a treatment-induced neuroendocrine phenotype [34]. Therefore, alternative biomarkers that can be used to efficiently monitor residual disease will have clinical utility. CTCs can potentially be used as a biomarker to detect metastasis and provide clinically relevant insight into the disease, but their prognostic potential remains elusive [35,36]. An association between CTC enumeration and endpoint tumour size was observed in our study, consistent with previous findings in other tumour types. It has been well established that CTC enumeration correlates with tumour size, progression-free survival (PFS) and overall survival (OS) in a variety of cancer types [37,38,39,40]. The large variation observed here in CTC enumeration across the different mouse blood samples is comparable to that observed between patient samples [41].

In metastatic castration-resistant prostate cancer, EMP of CTCs has been of particular interest due to the relationship between androgen deprivation and induction of EMP [42,43,44]. Consistent with this, EMP has predominantly been observed in castration-resistant prostate cancer [16,45]. Evidence suggests that drug resistance is associated with EMP characteristics, such as loss of E-cadherin and gain of N-cadherin, in prostate cancer [16,46,47].

Variation in cell size was observed in CTCs. While the majority of CTCs were approximately twice the size of PBMCs, some were comparable in size to PBMCs. “Small CTCs” have been reported previously, notably in castrate-resistant prostate cancer [48], and thus make size-based CTC isolation in prostate cancer challenging. Indeed, many CTC isolation strategies rely at least in part on CTCs having a larger size [11].

CTC clusters were observed in 9 of 11 samples. CTCs present in CTC clusters always appeared to be large, and any small cells observed in these clusters were not positive for any of our markers (VIM, KRT, SERPINE1, PSA) and are thus unlikely to be prostate cancer cells (Appendix A). Such heterogenous clusters, also referred to as heterotypic CTC clusters [11], consist of tumour cells along with non-malignant cells such as fibroblasts, white blood cells, platelets and endothelial cells [49]. The presence of non-malignant cells in CTC clusters has been associated with increased primary tumour growth rate and metastatic potential of CTCs [49,50,51,52,53]. Using blood from cancer patients and mouse models, it was shown that the presence of immune cells in CTC clusters increased cell proliferation rate [50]. In general, it is hypothesised that such CTC clusters increase their metastatic potential by “bringing their own soil”, which aids the establishment of new metastatic deposits [53]. However, more investigation is needed to understand the nature and consequences of interaction between these cells.

KRT^+^ or PSA^+^ tumour cells were often found on the periphery of CTC clusters, surrounding hybrid-like or mesenchymal-like cells (VIM^+^ or SERPINE1^+^). As CTC clusters have been previously shown to have 23–50-fold higher metastatic potential [54], mechanisms involved in holding the clusters together could be targeted to reduce the metastatic spread of cancer. Furthermore, as epithelial phenotype has been implicated in metastatic colonisation [17], the epithelial coating observed in CTC clusters may offer an advantage in this regard.

Molecular characterisation of CTCs allows us to understand the mechanisms that tumour cells use to survive in the circulatory systems and establish metastatic sites [55]. Our study explored the expression of a panel of genes primarily linked to EMP, a major candidate pathway for metastatic progression [17]. We identified various CTC subpopulations within our PDX sample cohort by staining CTCs for human VIM and KRT 8/18/19 (KRT). CTCs were either VIM^+^KRT^−^, VIM^−^KRT^+^ or VIM^+^KRT^+^; the latter we refer to as EMP hybrid cells. Four of 11 blood samples comprised of just one type of CTC (1x VIM^+^KRT^−^ only, 2x VIM^−^KRT^+^ only, 1x VIM^+^KRT^+^ only). In the remaining samples, varying proportions of the different types were present, although two samples did not have any detectable EMP hybrid cells (Table 1).

Use of human-specific RT-qPCR provides an opportunity to categorically assess CTCs in tumour xenograft models without confounding signals from non-tumour (murine) cell types [28]. The high RNA expression of both epithelial- and mesenchymal-associated genes in blood samples is consistent with these mixed CTC subpopulations, and could either correspond to hybrid CTCs or a mixture of epithelial and mesenchymal CTCs. Consistent with the identification of hybrid VIM^+^KRT^+^ cells detected using immunofluorescent staining (6/11 blood samples), a cluster of blood samples was observed within our cohort with a gene expression pattern consistent with a hybrid phenotype (Figure 5, indicated by black box). In particular, mesenchymal genes *FN1*, *INHBA*, *VIM*, *TNC*, *RRAS*, *SERPINE1*, *MSN*, *EMP3*, *LAMC2*, *NRP1* and *SNAI1* are highly expressed in this cohort. While analysing pooled expression data, it is important to consider the presence of heterogenous CTC populations, as shown by our CTC staining results. Additionally, single cell RNA-sequencing of the PDX-derived CTCs may be possible and would likely provide a more complete picture of EMP regulation, as it has in other systems [56,57]. Indeed, a single-cell sequencing study conducted on CTCs isolated from patients with metastatic prostate cancer provides evidence of the presence of hybrid phenotype in these cells [45]. Although 93% of these CTCs expressed *EPCAM*, consistent with our observations, a loss in epithelial phenotypic characteristics was often observed and most of these cells also expressed EMT-related genes. By contrast, a different single-cell RNA-sequencing study examining CTCs isolated from prostate cancer patients (a mixed group of localised, castrate-sensitive and metastatic castrate-sensitive and castrate-resistant patients) reported no evidence of EMT in CTCs [58]. However, downregulation of *KLK3* in comparison with their primary tumours was observed, consistent with our findings in all PDXs examined. This study also showed the upregulation of housekeeper genes *NONO* and *OAZ1* in CTCs, supporting our own data that these genes are not suitable to use as housekeeping genes in this setting.

Interestingly, the LuCaP96 model had no sample falling within the hybrid EMT gene expression sub-population. While hybrid VIM^+^KRT^+^ CTCs were present in all three blood samples from the LuCaP96 model, they were low in number (1, 3 and 14 CTCs per mL of blood). The LuCaP96 model also showed the least variation in gene expression between tumour and blood samples. Of the four PDXs examined, LuCaP96 is the only one derived from tumour at the primary site (prostate). By contrast, the other three PDXs were derived from metastatic deposits—BM18 and LuCaP105 from bone metastases and LuCaP70 from a liver metastasis [59,60]. It is possible that the clinical tumour site sampled to generate the PDX has influenced the ability of tumour cells to activate plasticity characteristics, although this needs to be examined in a larger cohort of PDXs. The tumours used to establish these PDX models all stain positive for PSA and KRT, and negative for VIM [59,60].

Several consistent differences in expression of EMP-associated genes between CTC and primary tumours were observed across the different PDX models. Higher levels of expression of mesenchymal-associated genes *SNAI1*, *SERPINE1*, *MSN*, *NRP1* and *LAMC2* were observed in CTCs compared to primary tumour in three of the four models (the exception being LuCaP96). By contrast, *CDH1* expression was unchanged in three of four PDX models and lower in LuCaP105 (where only 3/11 blood samples had any detectable signal for this gene). No significant changes in *VIM* (mesenchymal marker) or *EPCAM* (epithelial marker) were observed. The *CDH1* gene encodes E-cadherin, which is a transmembrane protein responsible for cell adhesion and polarity in epithelial cells [61]. Loss of or reduction in E-cadherin, as is the case during EMT in cancer cells, reduces cell–cell interactions and promotes cell motility and proliferation, and is frequently referred to as a hallmark of EMT [62]. EMP has quite often been defined by coordinated change in expression profiles of E-cadherin (epithelial marker) and vimentin (mesenchymal marker). VIM has been shown to promote metastatic spread in prostate cancer and plays a role in making tumour cells more invasive [63]. These results demonstrate that confining CTC analysis to a few markers can give an incomplete picture of cell state in the context of EMP.

Of particular interest is the relationship between Snail family members and E-cadherin. Snail-1, a transcriptional factor, silences *CDH1* expression, promoting EMT and cancer invasiveness [64]. Higher levels of Snail-1 expression were noted in the CTCs from all models except LuCaP96 (where a similar trend was observed but was not statistically significant). By contrast, *SNAI2*, was not detected in the majority of CTC samples despite it being readily detected in most tumours. Although substantial evidence is available validating the roles of *CDH1* and the Snail family in cancer invasiveness, a strong relationship between *CDH1* expression and *SNAI1/2* was not observed in the present study. Similarly, a study comparing colorectal cancer primary tumours and metastases has previously shown no significant association between *SNAI1* and *CDH1* [65]. Non-EMT-associated roles have been reported for factors such as Snail-1 [21], and it is noteworthy that Snail-1 can promote tumour progression without suppression of E-cadherin. It can also regulate the cell cycle and cell movement through alternative pathways and protect cancer cells against apoptosis [66,67], functions that would be conducive to CTC generation and survival.

*KLK3/PSA* is expressed by luminal epithelial cells in the prostate gland [68,69]. Earlier studies have shown that as tumour cells become more aggressive and motile, they reduce/lose expression of *KLK3* at the level of single cells [70,71]. Despite being often detected in CTCs across all our PDXs (BM18: 6/11, LuCaP70: 8/11, LuCaP96: 7/10, LuCaP105: 9/10), lower levels of *KLK3* were consistently observed in CTC samples compared to primary tumours (Appendix A). Cell staining revealed that PSA was often detected in cells within CTC clusters or in cells with a hybrid phenotype (VIM^+^ or SERPINE1^+^). Single PSA^+^ cells that were also negative for VIM and SERPINE1 were uncommon. *KLK3* is also known for its role in EMT induction in prostate cancer cell line PC-3 [62,72], and has been found to downregulate *CDH1* and regulate the cytoskeleton and cell migration process [62]. This heterogeneity of response of various prostate cancer cells to alterations in *KLK3* expression may explain why a lower level of *CDH1* in CTCs was only observed in one of four models, despite the consistent decrease in *KLK3* across all models, in the present study.

A strong increase in *SERPINE1* was also noted in three of the four models, with a similar trend in the fourth. *SERPINE1* has a well-established role in angiogenesis, tumour cell migration and proliferation [73,74,75], and has also been implicated in CTC extravasation from blood vessels to form metastatic deposits [76]. These previous reports are consistent with the relatively high expression of *SERPINE1* in the prostate cancer PDX CTCs, and was also a prominent finding in our earlier studies of breast cancer xenograft CTCs [28]. At the protein level, SERPINE1^+^ CTCs were detected in all PDX blood samples examined (*n* = 11). SERPINE1 plays a significant role in cancer cell dissemination through its role in cancer inflammation, resisting cell death through its anti-apoptotic activity, and it has been implicated in CTC extravasation with the help of neutrophil extracellular traps (NET) [77].

Our data also show a trend towards higher expression of *PPARGC1A* in CTCs compared to primary tumours, although the difference in expression was only statistically significant for BM18 and LuCaP105 models. This could indicate that the cells need more energy to survive in the unfavourable microenvironment of the circulation. *PPARGC1A* was found to be elevated in CTCs from a variety of experimental models [78], and has been shown to play a role in oxidative phosphorylation in breast and pancreatic cancer cells, helping them meet their growing demand for energy as they proliferate rapidly [78,79]. Furthermore, *PPARGC1A* upregulation has been shown as a determinant of cancer stem cell dependency on oxidative metabolism in pancreatic cancer [78].

CSCs are tumour cells with self-renewal capacity that exhibit resistance to therapy [80]. A common feature of CSCs in breast and other cancers is *CD24*^low^/*CD44*^high^ expression [81]. Studies regarding CSC markers in prostate cancer are ongoing and *CD44* and *CD24* are promising candidates [82]. *CD44* is known for its role in cancer cell migration and proliferation [83]. Consistent with its potential role in cancer cell dissemination, *CD44* was upregulated in CTCs in three of four models, while *CD24* was downregulated in two of four models (Appendix A).

*EPCAM* showed no significant change in expression between tumour and CTCs. Our data support the use of the epithelial marker *EPCAM* for CTC isolation and enumeration, as its expression level was unchanged in tumour cells in all four PDX models despite significant shifts in expression of many other genes. Notably, while both EPCAM^+^ and EPCAM^−^ CTCs were detected in metastatic prostate and breast cancer patients, only the EPCAM^+^ CTCs were associated with overall survival [84]. We conclude that functional studies of the most strongly and consistently overexpressed genes in CTCs, such as *SERPINE1*, *KRT20*, and *PPARGC1A*, are warranted to allow us to better understand their functional implications in metastasis. This may open a new window for the development of targeted therapies and personalised medicine. Interrogation of metastatic deposits using our 42-gene panel could help uncover their EMP status, and allow us to better understand which genes might be involved in the establishment of secondary tumours. However, none of these PDX models presented with detectable micro-metastasis within the study timeframe.

## 4. Methodology

### 4.1. Mouse Blood and Tissue Collection

The study examined four prostate cancer PDX models (BM18, LuCaP70, LuCaP96 and LuCaP105) [59,85]. All prostate cancer PDX models were grown as subcutaneous tumours in the lateral flank of male severe combined immune-deficient (SCID) mice as previously described, collected sequentially over a two-year period (except where collection was interrupted by COVID-19 restrictions), and under the approval of the relevant Animal Research Ethics Committees (TRI/QUT/370/17, QUT1800000289). All PDX work was carried out in accordance with the Australian National Health and Medical Research Council (NHMRC) guidelines for the care and use of laboratory animals. Blood (0.8–1.0 mL) was obtained from anaesthetised tumour-bearing mice by terminal draw via cardiac puncture. Pieces of the matched primary xenograft tumours were also collected and kept in RNAlater™ Stabilization Solution (Invitrogen, Waltham, MA, USA) overnight prior to storage at −80 °C until further processing.

### 4.2. Sample Processing

Blood samples were collected in 1.5 mL microfuge tubes containing 25 µL EDTA and further processed within 3 h of blood draw. To each sample, 4 mL red blood cell (RBC) lysis buffer (G-Biosciences, St. Louis, MO, USA) was added and incubated for 5 min at room temperature. Each sample was then spun at 400× *g* for 10 min at room temperature. After removing the supernatant, if the samples were going to be further processed for RNA analysis, they were resuspended in RNA lysis buffer from the ISOLATE II RNA Mini Kit (Bioline^®^, Sydney, Australia). The samples were stored at −80 °C until RNA was isolated using this kit according to the manufacturer’s protocol. For samples processed for immunocytochemistry (ICC) analysis, the cell pellets were resuspended in 600 µL Dulbecco’s modified Eagle’s medium (DMEM) containing 10% foetal bovine serum (FBS) (Gibco^TM^, Thermo Fisher Scientific, Waltham, MA, USA), and antibiotics penicillin and streptomycin (Gibco^TM^). Each sample was split onto 3 slides with 200 µL on each and cytospun at 150× *g* for 5 min. The slides were stored at −80 °C until immunostaining as described in Section 4.3 below.

Tumours were placed in RNA lysis buffer from the ISOLATE II RNA Mini Kit along with a 5 mm diameter stainless steel bead and homogenised at 30 Hz for 90 s using a TissueLyser II (Qiagen, Hilden, Germany). RNA isolation was then performed using the ISOLATE II RNA Mini Kit according to the manufacturer’s protocol.

### 4.3. Immunocytochemistry

Cytospun cells were fixed using 4% neutral buffered formalin (Sigma-Aldrich, St. Louis, MO, USA) for 15 min followed by permeabilization with 0.4% Triton X-100 for 10 min. The cells were then incubated in Background Sniper (Biocare Medical LLC, Pacheco, CA, USA) for 15 min, followed by 5% bovine serum albumin (BSA) (Sigma-Aldrich) for 30 min to block background. The cells were then incubated overnight with primary antibodies and for 2 h with secondary antibodies. Cytokeratins 8, 18 and 19 were detected by incubating with a cocktail of KRT antibodies (anti-KRT8 (HPA049866), anti-KRT18 (HPA001605) and anti-KRT19 (HPA002465) rabbit antibodies; all Sigma-Aldrich, USA) and secondary antibody labelled with Alexa Fluor^TM^ 647 (Invitrogen). Vimentin was detected by incubation with anti-vimentin (V9) human-specific primary antibody (cat no. 790-2917; Ventana Medical Systems, Oro Valley, AZ, USA) and secondary antibody labelled with Alexa Fluor^TM^ 488 (Invitrogen). SERPINE1 was detected by using PAI1 polyclonal antibody (cat no. PAI1-9077; Invitrogen) and secondary antibody labelled with Alexa Fluor^TM^ 568. PSA was detected using CONFIRM anti-prostate-specific antigen (PSA) rabbit polyclonal primary antibody (cat no. 760-2506; Ventana Medical Systems) and secondary antibody labelled with Alexa Fluor^TM^ 647 (Invitrogen). Diamidino phenyl indole (DAPI) was used as a nuclear stain and slides were then covered with Mowiol mounting solution followed by cover slipping. Cells were scanned using a 3DHISTECH Scan II Fluorescence Slide Scanner at 20× magnification (40 × 0.27 μm resolution), and images were collected and observed using CaseViewer 2.4 (3DHISTECH, Budapest, Hungary).

### 4.4. Molecular Analysis Using RT-qPCR of Mouse Xenograft Samples

RT-qPCR was used to examine gene expression in the blood and tumour samples. Primers were designed to detect human, but not murine, transcripts (human-specific; Appendix A). Due to the relatively low concentration of human RNA in the mouse white blood cell fractions, emphasis was placed on maximal input; 10 µL of eluted RNA out of a total of 40 µL was added to each cDNA synthesis reaction [28]. By contrast, since the tumour samples were primarily comprised of human RNA and were highly concentrated, they were diluted to a working concentration of 20 ng/µL and 10 µL was added to each reaction. The results were normalised to the RPL32 gene to control for varying inputs. A blinded analysis was performed, and included a negative control sample (blood drawn from a non-tumour bearing mouse) in the sample cohort. This sample identification number was not disclosed until the end of the pre-screen and was used to confirm that false positives were not detected in these experiments.

### 4.5. cDNA Synthesis from RNA Samples

Reverse transcription (RT) was performed using the SuperScript™ IV First-Strand Synthesis System (Invitrogen™, USA). To each 10 µL of sample, 4 µL of master mix-1 containing 2 µL RT Primer pool (8 mM) and 2 µL 10 mM dNTP was added. The samples were heated in a PCR machine (GeneAmp^®^ PCR System 9700, Applied Biosystems, Waltham, MA, USA) to 65 °C for 5 min and then cooled to 55 °C, and brought to a hold. A second master mix, containing 4 µL 5x First Strand Buffer, 1 µL 0.1 M DTT, 1 µL Ribonuclease Inhibitor and 0.1 µL SS IV reverse transcriptase enzyme for each reaction, was placed in the machine for 1 min to equilibrate it to 55 °C, after which 6.1 µL of it was added to each tube of sample with master mix-1. The reaction was left at 55 °C for 1 h, and then directly heated to 85 °C for 5 min to inactivate the enzyme after which the block was left at 4 °C until the sample was collected.

### 4.6. cDNA Precipitation

To each RT product, 2 µL 20 mg/mL glycogen was added, followed by 180 µL of a sodium acetate and ethanol solution (5 µL 2M sodium acetate and 175 µL 100% ethanol per reaction). The samples were kept at −20 °C for 1 h and then spun at 11,000× *g* for 1 h at 4 °C. The pellet was washed in 200 µL of 75% ethanol and spun again at 13,000× *g* for 15 min at 4 °C. After carefully discarding the supernatant, the pellet was air dried for 10 min, resuspended in 10 µL of nuclease-free water, and stored 4 °C until further processed.

### 4.7. Pre-Amplification

Blood samples were pre-amplified using a set of human-specific outer primers (Appendix A) for 15 cycles. For each 10 µL sample, a master mix containing 12.5 µL of 2× SYBR premix, 1.25 µL RNase-free water and 1.25 µL of 4 µM outer primer pool (25×) was used. After pre-amplification, the reaction products and tumour samples were both diluted 1:125 in nuclease-free water.

### 4.8. Tandem Nested RT-qPCR

SYBR Green MasterMix and custom primers were used to perform a qPCR run on all samples including positive and negative controls using a Life Technologies ViiA™ 7 Real-Time PCR System (USA) in a 384-well format. To each 5 µL of reaction, 5.4 µL of mix was added, containing 5.21 µL 2× PreMix and 0.21 µL 10 µM inner primer pool. While negative controls for most genes exhibited no amplification signal, where Ct values were observed, the data were adjusted to only include values that were 2 Ct values less than the negative control to account for background signal.

### 4.9. Testing Specificity of RPL32 Primer Set

The *RPL32* primers were tested using the 4T1 mouse mammary cell line [86] from Fred Miller, Michigan Cancer Foundation, USA and the MDA-MB-231 human breast cancer cell line from the American Type Culture Collection (Manassas, VA, USA). A complete RT-qPCR was performed on 1ng of RNA from each cell line.

### 4.10. Estimation of Total Number of CTCs per Mouse

As a typical mammalian cell has been estimated to containing 10–30 pg of RNA [87], we used the lower end of this range (10 pg) to estimate CTC number. Therefore, 10 ng of RNA from a known positive control sample (PDX LuCaP141 tumour piece, >99% tumour purity) was used in the reaction to represent 1000 cells worth of RNA The *RPL32* cycle threshold (Ct) value for the positive control was then used to estimate the total number of CTCs in the reaction volume for each sample. Each sample was then corrected for the proportion of extracted RNA used in each reaction (one-quarter) and estimated total mouse blood volume (10% volume/body weight) [88].

### 4.11. Statistical Analysis

Gene expression (RT-qPCR) results were analysed using Microsoft^®^ Excel, GraphPad Prism 8.3.0. and Morpheus (https://software.broadinstitute.org/morpheus/, accessed 14 February 2020). Ct values were normalised to housekeeper gene, RPL32. False discovery was determined using the Two-stage linear step-up procedure of Benjamini, Krieger and Yekutieli, with Q = 5%. Each gene was analysed individually without assuming a consistent SD. Correlation analysis was performed using non-parametric Spearman’s correlation and a 95% confidence interval.

## 5. Conclusions

Molecular characterisation of CTCs from PDX models can inform understanding of the mechanisms behind tumour cell dissemination and help us develop new ways to monitor disease progression. This approach provides a platform for pre-clinical testing of new therapies and an opportunity to identify novel therapeutic targets that can pave the way for personalised medicine. Caution should be taken while interpreting pooled CTC analysis due to high CTC heterogeneity across the EMP axis. Our study indicates changes in epithelial and mesenchymal gene expression in CTCs as compared to tumour samples. Rather than epithelial gene expression changes being opposite to the direction of mesenchymal changes, in many cases, both were elevated in CTCs. Several specific gene expression patterns were consistent among all models despite having different clinical starting material, while other changes were model-specific. These gene signatures could be attributed to hybrid CTCs or the presence of both epithelial and mesenchymal CTCs. In the context of the demonstration of hybrid CTCs using immunofluorescence staining, this study contributes to the growing recognition of the importance of hybrid EMP states in metastasis [17]. Furthermore, our findings suggest caution should be taken while using size-based CTC isolation technologies as CTCs vary significantly in cell size, as observed in earlier studies as well [89,90].

## Figures and Tables

**Figure 1 cancers-13-02750-f001:**
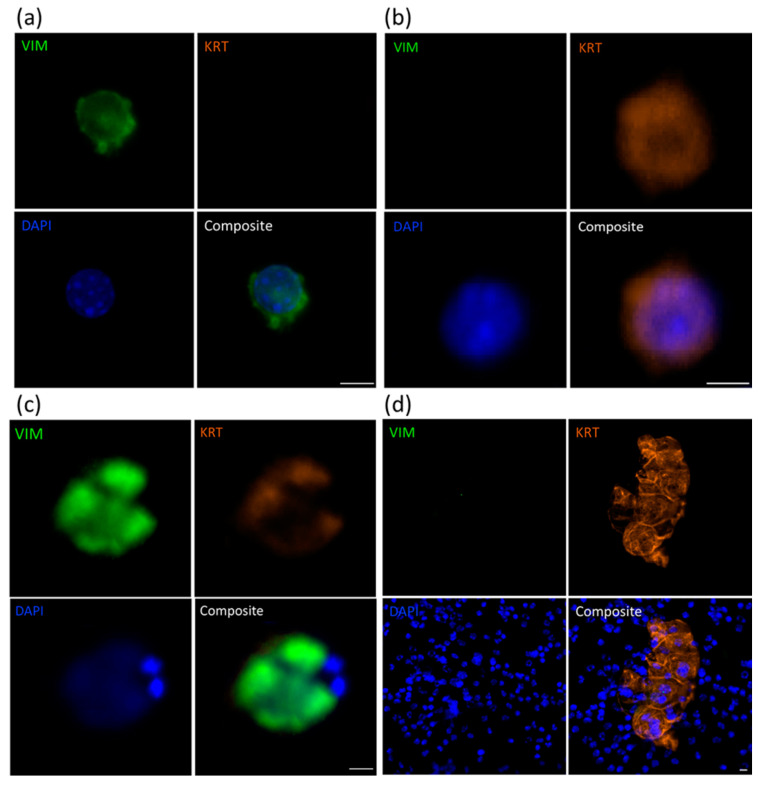
Immunofluorescence staining for vimentin (VIM) and cytokeratins 8/18/19 (KRT) in PDX-derived CTCs. Representative images of LuCaP70 CTCs staining for (**a**) VIM only, (**b**) KRT only or (**c**) both VIM and KRT; (**d**) KRT-positive CTC cluster surrounded by murine PBMCs. Scale bar denotes 5 µm.

**Figure 2 cancers-13-02750-f002:**
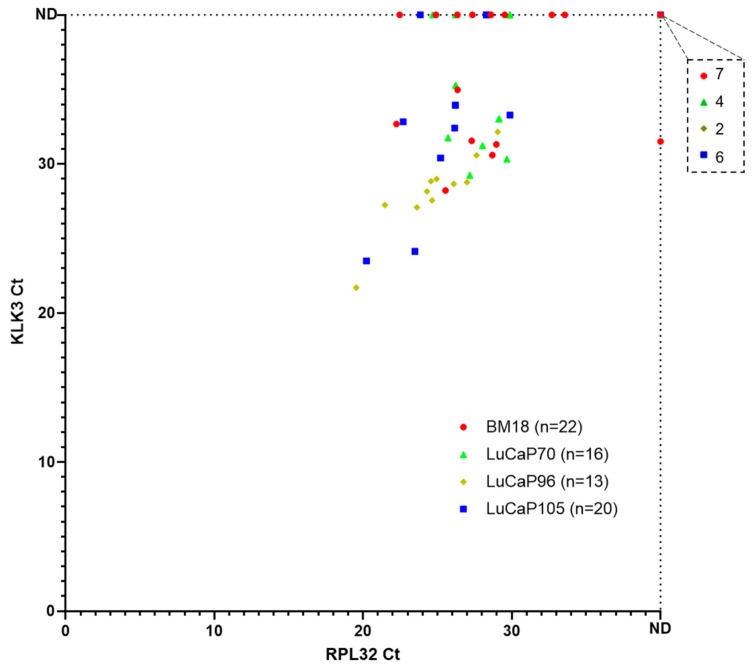
Expression of *RPL32* and *KLK3* in blood samples collected from prostate cancer PDX-bearing mice. Scatter plot of *RPL32* versus *KLK3* cycle threshold (Ct) values for nested RT-qPCR pre-screen results from 4 prostate cancer PDX models. Each dot corresponds to one mouse blood sample (*n* = 1 per mouse) and smaller Ct values correspond to higher gene expression. The numbers of samples that were negative for both RPL32 and KLK3 are indicated on the top right breakout box. ND, not detected.

**Figure 3 cancers-13-02750-f003:**
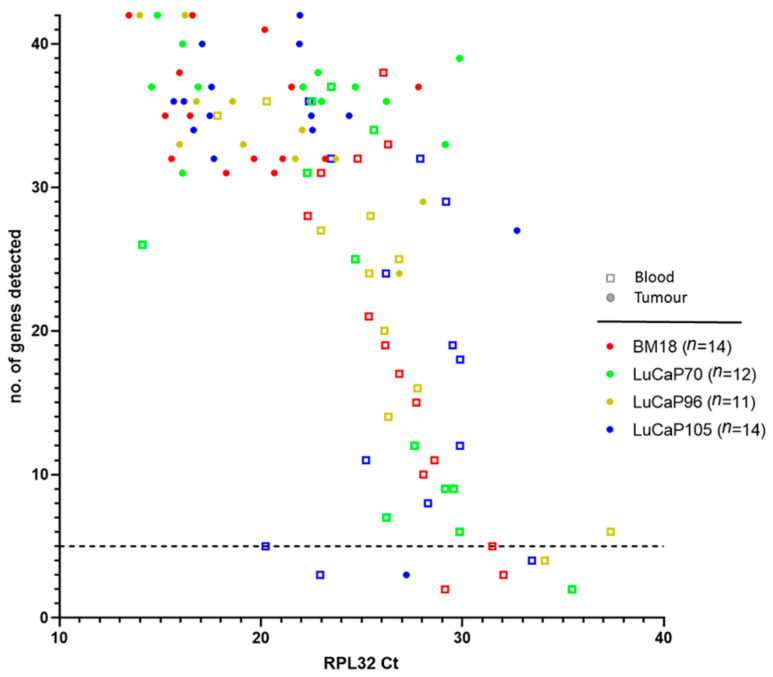
The number of genes detected in blood and tumour samples decreased sharply with lower expression of housekeeping gene *RPL32*. The complete 42-gene human-specific nested RT-qPCR assay was performed on *RPL32*-positive blood samples and their paired tumours. The number of genes detected in each sample was plotted against their raw cycle threshold (Ct) value of *RPL32*. Smaller Ct values correspond to higher gene expression. *n* denotes number of blood and tumour pairs for each PDX. Dotted line indicates 5 genes detected.

**Figure 4 cancers-13-02750-f004:**
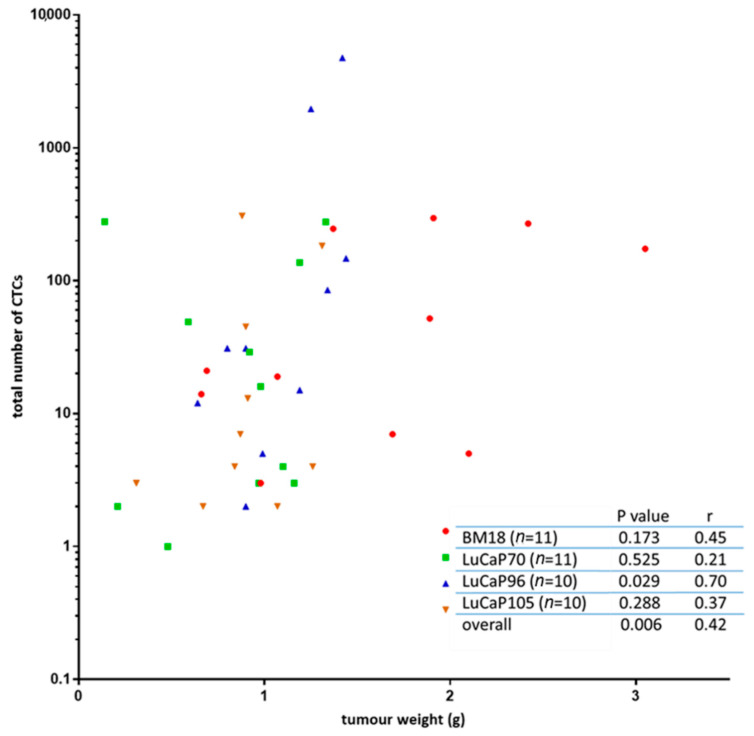
Relationship between tumour weight and number of circulating tumour cells (CTCs). The tumour weight for each mouse was compared to the estimated total number of CTCs in that mouse’s whole blood volume. Limited correlation was identified between the two variables using a non-parametric Spearman correlation and a 95% confidence interval for each model individually, and overall, a significant correlation was observed. Each symbol represents a sample from an individual mouse (*n* = 1 per mouse).

**Figure 5 cancers-13-02750-f005:**
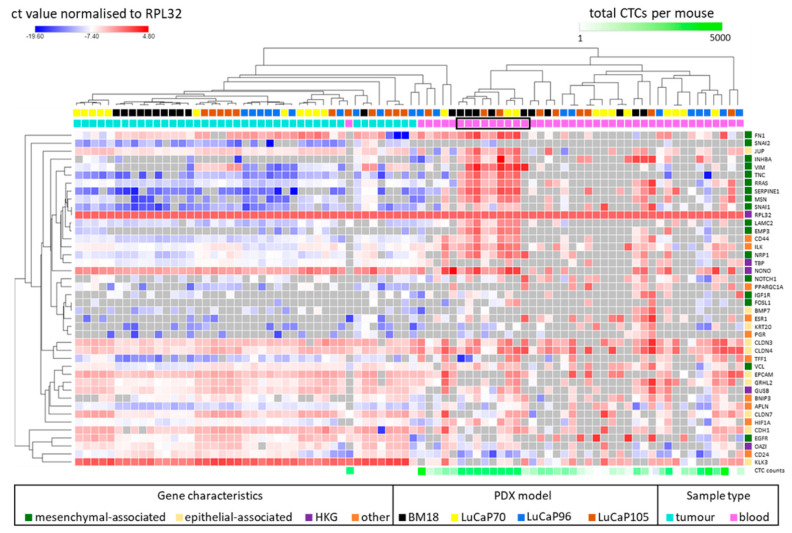
Heat map showing 42 gene panel expression profiles in blood and matched tumour samples as Ct values normalised to *RPL32*, with global normalisation. Hierarchical unsupervised clustering was performed using one minus Pearson correlation. For each sample, the PDX model is shown above the heat map, followed by colour coding to indicate tumour (teal) and blood (pink) biospecimens. The final row indicates estimated circulating tumour cell (CTC) counts for each blood sample. Black block indicates cluster of blood samples showing evidence of a hybrid phenotype. Genes are categorised as mesenchymal-associated, epithelial-associated, potential HKG (*GUSB*, *TBP*, *OAZ1* and *NONO*), and others (which includes HR- (*ESR1*, *PGR* and *TFF1*), CSC- (*CD24* and *CD44*), hypoxia- (*APLN*, *HIF1A* and *BNIP3*), metabolism- (*PPARGCIA*), and anoikis- (*ILK*) associated genes). HR, hormone receptor; CSC, cancer stem cell; HKG, housekeeper gene.

**Figure 6 cancers-13-02750-f006:**
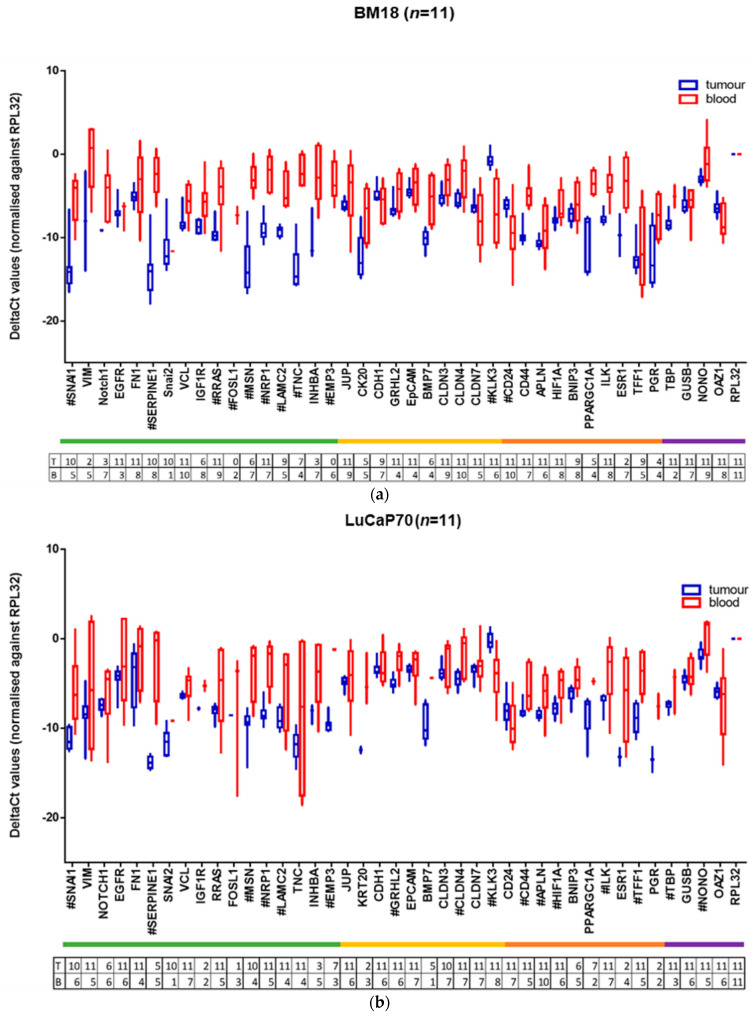
(**a**–**d**): Box and whisker plots for tumour (T) vs. blood (B) samples of the different prostate PDX grafted mice. ΔCt values are normalised to housekeeper gene *RPL32*. Significant differences in gene expression values between CTCs and tumours were identified using the Benjamini, Krieger and Yekutieli two-stage linear step-up procedure to control the false discovery rate (Q = 5%, *p* < 0.05), and are marked with “#”. Numbers of samples included in each data point indicated in table below each graph. (Each tumour (T)/blood (B) pair is from a different mouse.)

**Figure 7 cancers-13-02750-f007:**
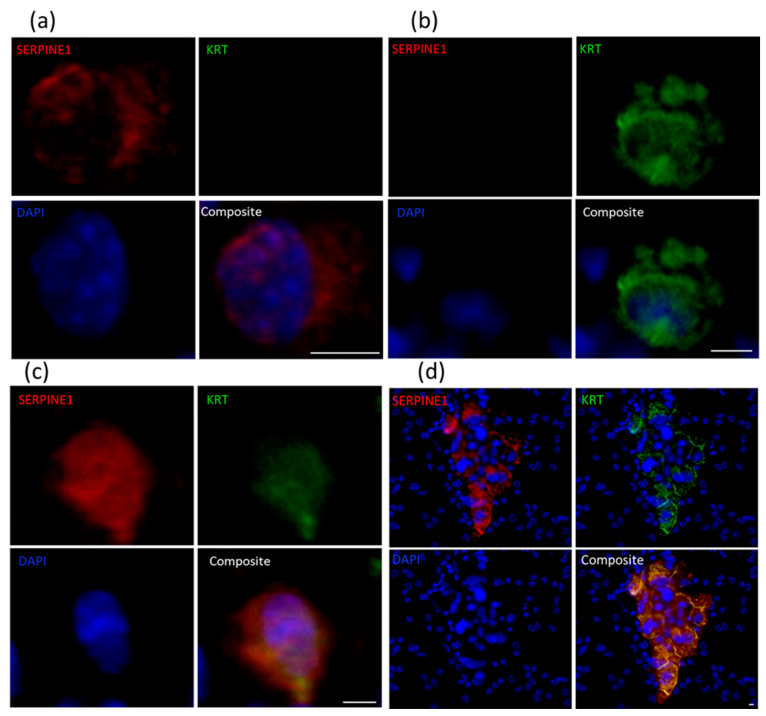
Immunofluorescent staining for SERPINE1 (red) and KRT 8/18/19 (green) in PDX-derived CTCs. Representative images using LuCaP96 mouse blood of single CTCs staining positive for (**a**) SERPINE1 only, (**b**) KRT only or (**c**) both SERPINE1 and KRT. (**d**) Representative images using LuCaP70 mouse blood of SERPINE1^+^KRT^+^ CTC cluster surrounded by murine PBMCs. Scale bar denotes 5 µm.

**Figure 8 cancers-13-02750-f008:**
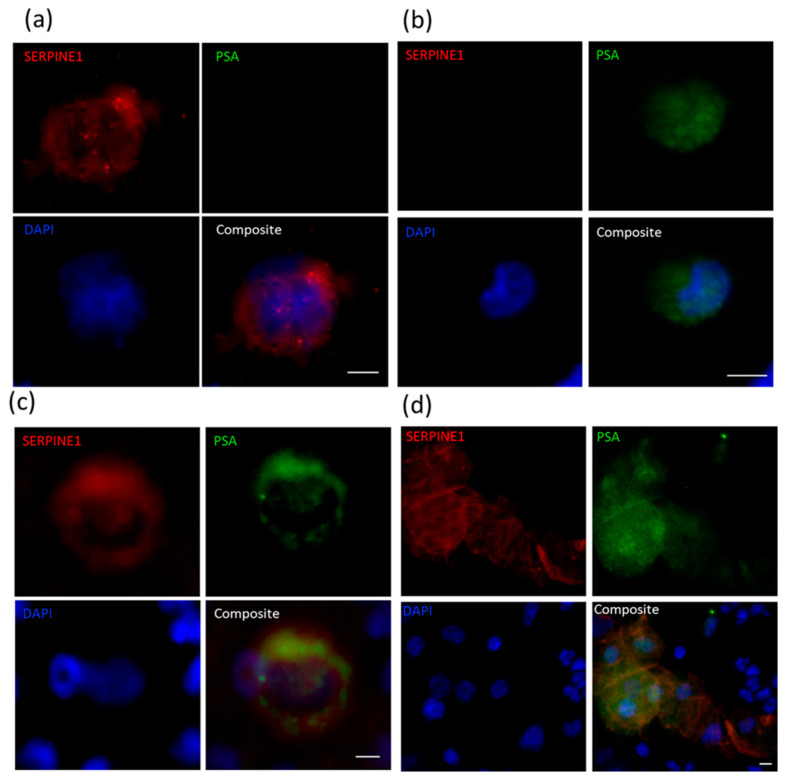
Immunofluorescent staining for SERPINE1 and PSA in PDX-derived CTCs. Representative images using PDX mouse blood of CTCs staining positive for (**a**) SERPINE1 only (LuCaP96), (**b**) PSA only (LuCaP70) or (**c**) both SERPINE1 and PSA (LuCaP96), and (**d**) CTC cluster composed of both SERPINE1^+^PSA^−^ cells and SERPINE^+^PSA^+^ cells surrounded by murine PBMCs (LuCaP70). Scale bar denotes 5 µm.

**Table 1 cancers-13-02750-t001:** Circulating tumour cell vimentin/keratin immunofluorescence cytology for PDX blood samples.

PDX ID	Total Detected (CTCs/mL)	Clusters (Clusters/mL)	VIM^+^KRT^−^ (%)	VIM^−^KRT^+^ (%)	VIM^+^KRT^+^ Hybrid (%)
BM18: 20-3	4	0	50	50	0
BM18: 20-28	4	0	0	25	75
BM18: 20-132	400	38	29	49	22
LuCaP70: 20-6	901	5	2	98	0
LuCaP70: 20-105	5	0	0	100	0
LuCaP70: 20-112	1294	100	45	52	3
LuCaP96: 20-7	3	0	0	0	100
LuCaP96: 20-33	114	13	69	18	13
LuCaP96: 20-125	21	0	84	11	5
LuCaP105: 19-183	7	0	0	100	0
LuCaP105: 20-102	1	0	100	0	0

CTC, circulating tumour cell; KRT, keratin 8/18/19; PDX, patient-derived xenograft; VIM, vimentin.

**Table 2 cancers-13-02750-t002:** Summary of proportion of positive samples for each PDX model.

PDX	Total Samples (*n*)	Positive Samples ^#^ (*n*)	Percentage Positive (%)
BM18	22	11	50
LuCaP70	16	11	69
LuCaP96	13	10	77
LuCaP105	20	10	50

^#^ Samples with at least 5 detectable genes were considered positive.

**Table 3 cancers-13-02750-t003:** Summary of gene expression differences between CTC and primary tumour samples (*p* values shown) across PDX models.

GENE	BM18	LuCaP70	LuCaP96	LuCaP105
SNAI1	0.0000985	0.0011992	NS	0.0000715
VIM	NS	NS	NS	NS
NOTCH1	0.0388021	NS	NS	NS
EGFR	NS	NS	NS	NS
FN1	NS	NS	NS	NS
SERPINE1	0.0000001	0.0005100	NS	0.0003550
SNAI2	NS	NS		NS
VCL	0.0011664	NS	NS	NS
IGF1R	0.0152763	NS	NS	NS
RRAS	0.0001359	NS	NS	0.0000048
FOSL1		NS	NS	NS
MSN	0.0000247	0.0008278	NS	0.0068664
NRP1	0.0000003	0.0000273	NS	0.0184160
LAMC2	0.0000368	0.0225527	NS	0.0036031
TNC	0.0000312	NS	NS	0.0000017
INHBA	0.0120652	NS	NS	NS
EMP3		0.0000003	NS	NS
JUP	NS	NS	NS	0.0164246
KRT20	0.0303967	NS	NS	NS
CDH1	NS	NS	NS	0.0000830
GRHL2	0.0218429	0.0001708	NS	NS
EPCAM	NS	NS	NS	NS
BMP7	0.0070227	NS	NS	NS
CLDN3	0.0044251	NS	NS	NS
CLDN4	0.0019099	0.0008343	0.0004490	NS
CLDN7	NS	NS	NS	0.0083468
KLK3	0.0001113	0.0001525	0.0009885	0.0000037
CD24	0.0029071	NS	NS	0.0135173
CD44	0.0000004	0.0077010	NS	0.0000067
APLN	NS	0.0081341	NS	0.0310010
HIF1A	0.0190586	0.0027857	NS	NS
BNIP3	NS	NS	NS	NS
PPARGC1A	0.0073689	NS	NS	0.0000008
ILK	0.0000058	0.0175717	NS	0.0083933
ESR1	0.0273697	NS	NS	NS
TFF1	NS	0.0001196	0.0001279	
PGR	0.0759697	NS		NS
TBP	0.0009111	0.0183997	NS	0.0010770
GUSB	NS	NS	NS	NS
NONO	0.0160096	0.0133378	NS	NS
OAZ1	0.0226327	NS	NS	NS

Red boxes depict upregulation (higher in CTCs compared to primary tumour); blue boxes depict downregulation (lower in CTCs compared to primary tumour); white boxes have either no significant change (NS; *p* > 0.05) or insufficient data for analysis (blank). Significant differences in gene expression values between CTCs and tumours were identified using the Benjamini, Krieger and Yekutieli two-stage linear step-up procedure to control the false discovery rate (Q = 5%, *p* < 0.05).

**Table 4 cancers-13-02750-t004:** Summary of circulating tumour cell SERPINE1/KRT immunofluorescence cytology for all PDX blood samples.

PDX ID	Total Detected (CTCs/mL)	Clusters (Clusters/mL)	SERPINE1^+^ KRT^−^ (%)	SERPINE1^−^ KRT^+^ (%)	SERPINE1^+^ KRT^+^ Hybrid (%)
BM18: 20-3	11	0	0	60	40
BM18: 20-28	23	0	9	91	0
BM18: 20-132	694	68	0	0	100
LuCaP70: 20-6	779	29	0	100	0
LuCaP70: 20-105	0	0	0	0	0
LuCaP70: 20-112	950	56	0	1	99
LuCaP96: 20-7	26	0	0	11	89
LuCaP96: 20-33	178	13	14	28	58
LuCaP96: 20-125	82	6	38	19	43
LuCaP105: 19-183	0	0	0	0	0
LuCaP105: 20-102	0	0	0	0	0

CTC, circulating tumour cell; KRT, keratin 8/18/19, PDX, patient-derived xenograft.

**Table 5 cancers-13-02750-t005:** Summary of circulating tumour cell VIM/SERPINE1/PSA immunofluorescence cytology for all PDX blood samples.

PDX ID	Total Detected (CTCs/mL)	Clusters (Clusters/mL)	SERPINE1^+^PSA^−^ (%)	VIM^+^PSA^−^ (%)	SERPINE1^−^PSA^+^ or VIM^−^PSA^+^ (%)	SERPINE1^+^PSA^+^ or VIM^+^PSA^+^ Hybrid (%)
BM18: 20-3	71	4	13		75	13
BM18: 20-28	8	1		13	0	88
BM18: 20-132	635	24	35		3	62
LuCaP70: 20-6	13	0		70	0	30
LuCaP70: 20-105	8	0	88		0	13
LuCaP70: 20-112	1166	70	3		49	48
LuCaP96: 20-7	17	0	33		0	67
LuCaP96: 20-33	31	4	32		0	68
LuCaP96: 20-125	24	2	23		5	73
LuCaP105: 19-183	9	1	75		13	13
LuCaP105: 20-102	3	1		67	0	33

Grey box, staining not performed for that antibody combination. CTC, circulating tumour cell; PDX, patient-derived xenograft; PSA, prostate-specific antigen; VIM, vimentin.

## Data Availability

Data supporting these results has been deposited in QUT’s research data repository Research Data Finder (RDF) (https://researchdatafinder.qut.edu.au/, accessed 30 April 2021).

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
