# Peer review of "Diversity of Epithelial-Mesenchymal Phenotypes in Circulating Tumour Cells from Prostate Cancer Patient-Derived Xenograft Models"

_cancers, 2021, doi:10.3390/cancers13112750_

Round 1
Reviewer 1 Report
Sara Hassan and colleagues investigate here the phenotype of so-called circulating tumor cells (CTCs) in a manuscript entitled „Dysregulated epithelial-mesenchymal plasticity in circulating tumour cells from prostate cancer patient-derived xenograft models“. The authors utilized here four different patient-derived xenograft mouse models for prostate cancer combined with immunofluorescence staining and nested quantitative Real-Time PCR approaches to assess the expression of epithelial or mesenchymal specific marker in CTCs and their corresponding primary tumors.
The diagnostic of cancers often requires invasive techniques for biopsies representing a non-negligible burden for patients and wounding the lesion with unpredictable consequences. Liquid biopsies and CTC-based diagnostic methods may therefore represent potential less invasive attractive alternatives. The topic of this study is therefore overall of interest.
The present manuscript, unfortunately, presents a series of flaws making it not appropriate for publication at Cancers. Among others:
- The aims, findings, and take-home message of this work are not clearly stated.
- The description of the results is often very confusing and lacks proper interpretation.
- The experimental approach mainly relies on quantitative real-time PCR of whole blood samples and does not allow to distinguish between hybrid epithelial-mesenchymal states and mixtures of CTCs of epithelial and mesenchymal differentiation.
- The focus on “epithelial-mesenchymal plasticity” stated in the title is inappropriate as the study does not investigate changes of differentiation states of the CTCs (plasticity reflects the ability to switch between differentiation states).
- The language quality does not meet the requirements for a publication in Cancers.
Minor comment: standard gene annotation nomenclature rules are not respected.
Author Response
We have used this feedback to improve the manuscript. A detailed description of our response follows.
The aims, findings, and take-home message of this work are not clearly stated.
We have more clearly stated and expanded upon our statement of aims at the end of the introduction section (lines 123-125):
“The current study aims to expand our understanding of prostate cancer by studying PDX-derived CTCs using both molecular PCR-based and cell staining approaches, with a particular focus on EMP characteristics that may regulate tumour cell dissemination.”
Having re-read the conclusions section, we can find nothing that would warrant the Reviewer’s statement that the findings and take-home message are not clearly stated. We have shown this manuscript to colleagues not associated with the publication and asked for advice regarding the clarity of the aims, findings and take-home message, and they have all reported that they understood what was being stated. We do note that in accordance with the prescribed layout for Cancers, the conclusions section is separated from the discussion section by the methodology section. As such, it is possible that Reviewer 1 may have missed the conclusions section entirely.
The description of the results is often very confusing and lacks proper interpretation.
We accept that the results section represents a considerable amount of data, which can be difficult to follow. Following careful review and reflection we have made the following corrections/adjustments to improve the results section:
The hyphens describing the positive staining status of CTCs were formatted such that they looked like negative symbols. This has been corrected: “VIM and/or cytokeratin (KRT) -positive cells” (lines 129-130). We have also used superscript negative and positives throughout the manuscript to avoid this confusion.
For Tables 1, 4 and 5, we have modified the column headings to make it clearer what the CTCs stained positively/negatively for, changed “cells/mL” to “CTCs/mL” and for clusters changed the units to “clusters/mL”.
We have removed “in blood samples” from a sentence about CTC populations as it was superfluous. (Line 137)
For the staining characteristics of the CTCs, we have rephrased our descriptions to make it clearer what cell populations were present, i.e. “only VIM+ cells”, rather than “VIM only positive cells”. (Line 138, and following)
On line 144, we have removed “CTCs” from the sample description in parentheses, as it is describing the proportion of samples, not CTCs.
We have added in “tandem-nested” to the description of the RT-qPCR assay on line 164 of the results, to better convey the high sensitivity/specificity of the technique.
We have corrected a comma to a semicolon on line 176.
For Figure 2, we have replaced the figure with a revised version that includes an insert (breakout box) to indicate the number of samples that were negative for both RPL32 and KLK3 expression, as these were obscured given multiple data points were at the same location on the X-Y plot.
The sentence at the end of a paragraph (line 243) has been deleted, as it repeated part of what had already been stated previously in the paragraph.
In the legend for Figure 5, “with global normalisation” has been moved from a sentence about the hierarchical clustering to a statement about the heatmap itself, as this is what it better relates to. Also, “circulating tumour cell (CTC) counts” has been changed to “estimated circulating tumour cell (CTC) counts”, as the number of CTCs are not directly measured using this approach.
On line 318, “any expression” has been modified to “any detectable expression”, to better reflect that the assay has sensitivity limits.
We have changed “the majority of the CTC population in 5 of 6 samples” to “the majority of the CTC population in 4 of 6 samples”, as one sample, while still representing the largest subpopulation, wasn’t the majority of the CTCs. (line 362)
On line 364, “SERPINE1-KRT. cells” has been corrected to “SERPINE1-KRT+ cells”, and the molecular phenotypes being discussed defined more clearly.
On line 367 a typographical error “id” has been corrected to “did”.
The experimental approach mainly relies on quantitative real-time PCR of whole blood samples and does not allow to distinguish between hybrid epithelial-mesenchymal states and mixtures of CTCs of epithelial and mesenchymal differentiation.
We have employed both human-specific, tandem nested reverse-transcriptase quantitative PCR and human-specific immunofluorescent staining approaches. While we clearly state the limitations of pooled approaches (below), which affect almost all studies, the presence of EMP-hybrid cells is confirmed by our CTC staining results. The issue identified by the reviewer of not being able to distinguish between states is directly addressed by eth following statements in the manuscript:
“The high RNA expression of both epithelial- and mesenchymal-associated genes in blood samples is consistent with these mixed CTC subpopulations, and could either correspond to hybrid CTCs or a mixture of epithelial and mesenchymal CTCs. While analysing pooled expression data, it is important to consider the presence of heterogenous CTC populations, as shown by our CTC staining results.” (lines 460-463)
“Caution should be taken while interpreting pooled CTC analysis due to high CTC heterogeneity across the EMP axis.” (lines 698-899)
The focus on “epithelial-mesenchymal plasticity” stated in the title is inappropriate as the study does not investigate changes of differentiation states of the CTCs (plasticity reflects the ability to switch between differentiation states).
We accept this point and have modified the title to “Diversity of epithelial mesenchymal phenotypes in circulating tumour cells from prostate cancer patient derived xenograft models” which better represents our study.
The language quality does not meet the requirements for a publication in Cancers.
We are perplexed by this assessment. Between us we have published many high quality English language publications and reviewed countless manuscripts, and consider the English language quality of this manuscript to be of a reasonable standard. We have revised the manuscript to improve readability as indicated by the tracked changes throughout.
Minor comment: standard gene annotation nomenclature rules are not respected.
We have identified and corrected 2 instances where an abbreviated form of a gene symbol was used (L32 instead of RPL32 on line 275 and 306) and an oversight where we had used the non-standard EpCAM rather than EPCAM (lines 574-575). Throughout the body of the text we have changed the formatting of the gene symbols so that they are now in italics to conform with standard convention. The manuscript refers to the KLK3 (Kallikrein Related Peptidase 3) protein as PSA (prostate specific antigen), as this is what it is more commonly known as, and we indicate this in the abstract (line 70).
Reviewer 2 Report
The authors have answered my previous concerns and therefore I recommend the manuscript for publication.
Author Response
No Comments and Suggestions for Authors
“The authors have answered my previous concerns and therefore I recommend the manuscript for publication.”
Reviewer 3 Report
The present study analyzes a quite interesting and contemporary topic, adding a new perspective regarding CTCs and prostate cancer progression. However, in my opinion, the structure of the manuscript does not help the reader to follow neither the experimental workflow nor the findings. It is not clear how many samples were used in experiments and why this number varies from experiment to experiment. Therefore, it is difficult to understand the conclusions of the study. Should the following issues be addressed, I believe that it merits publication in Cancers:
Methodology:
- I believe that a figure explaining the experimental workflow would be helpful, since the readers could better understand the approach, which was followed, and the results, as well.
- Could the authors please explain their choice to use RPL32 gene as reference gene?
- The section in methodology in which Immunocytochemistry is described should be transferred in an individual paragraph and explained in more detail.
- Did the authors use a calibrator for the real-time experiments?
- How many times were the experiments conducted?
Results:
- It is not clear which were these 11 representative blood samples, which are mentioned in line 128.
- The molecular differences between xenografts should be discussed more thoroughly and differences among the gene expression in each xenograft should be pointed out.
- In line 165, it is mentioned that 85 samples were analyzed, but I do not understand which samples were these.
- Do the authors have a potential explanation why the number of CTCs was so different among the blood samples?
- The authors should emphasize more in explaining their selection of the 42 gene panel.
- This finding could be more analyzed: “The remainder of blood samples did not show bias towards an epithelial or mesenchymal state, nor were they similar to their matched primary tumours.”
Discussion:
- The authors could discuss the limitations and future perspectives of their study in the “Discussion”.
Author Response
We have used Reviewer #;s comments to further improve the manuscript.
"Should the following issues be addressed, I believe that it merits publication in Cancers."
Methodology:
1- I believe that a figure explaining the experimental workflow would be helpful, since the readers could better understand the approach, which was followed, and the results, as well.
We thank the reviewer for this helpful suggestion. We have made a new figure for the experimental workflow clearly showing the number of samples at each stage and the experimental processes involved (Supplementary figure 1).
2- Could the authors please explain their choice to use RPL32 gene as reference gene?
RPL32 was used to normalise the data as it was the most frequently detected of the predefined 4 house-keeping genes measured. The expression of other housekeeper genes varied quite significantly between samples so might add a bias in our results. We have also assessed RPL32 both in the Klijn 675 cell line data set (Nat Biotechnol 2015 PMID: 25485619) and the Nguyen 2018 Nature Communications single cell RNA-seq (PMID: 29795293) dataset, and found it to be one of the most effective normalising genes, based on either minimum variation across all samples (675 cell lines) or minimizing the standard deviation of other genes across all individual cells. Similarly, Chen and colleagues (Prostate 2013, PMID: 23280481) assessed possible housekeeping genes ACTB, GAPDH, and UBB and found that expression levels of ACTB and GAPDH were less stable and weaker among different CTCs compared to UBB in metastatic prostate cancer patients and used a single gene (UBB) for normalisation. We can see from our analysis in figure 5 that the other housekeeping gene sometimes do vary significantly between tumour and blood samples. For these reasons, we have chosen RPL32 alone to normalise the data.
3- The section in methodology in which Immunocytochemistry is described should be transferred in an individual paragraph and explained in more detail.
Sample processing (Section 4.2, commences line 596) and immunocytochemistry (Section 4.3, commences line 613) are two clearly identified separate sections in the methodology. The methods section for immunocytochemistry is 17 lines long and already includes more information than is typical for published methodologies. It may be that the reviewer read “processed for immunocytochemistry (ICC) analysis” in the sample processing section, and missed that there was a separate section detailing this aspect of the methodology – to address this we have added a reference to this section (“as described in Section 4.3 below”, line 608).
4- Did the authors use a calibrator for the real-time experiments?
For the human-specific, tandem nested reverse-transcriptase quantitative PCR experiments, a known positive control sample was included in all experimental batches, and was used in the estimation of total number of CTCs per mouse, as described in the methods (lines 678-685).
5- How many times were the experiments conducted?
All blood and tissue samples were from mice sequentially collected during the routine maintenance of prostate cancer PDX models, collected over a two-year period (with some interruptions due to COVID19 restrictions). The relevant methods section (lines 588-589) has been modified to include these details. RNA analysis was performed in batches, as described in the results section (line 163).
Results:
1- It is not clear which were these 11 representative blood samples, which are mentioned in line 128.
The 11 representative samples, as shown in table 1, are 3 BM18, 3 LuCaP70, 3 LuCaP96 and 2 LuCaP105. We have now added this information to our new figure (Figure S1) describing the experimental workflow, to make this clearer for the reader.
2- The molecular differences between xenografts should be discussed more thoroughly and differences among the gene expression in each xenograft should be pointed out.
While the focus of our study was to compare the difference in gene expression between tumours and CTCs in relation to their EMP status, rather than different PDX tumours, we have included a new paragraph discussing the differences between gene expression in the 4 PDX models to address this comment (lines 344-352).
“Gene expression for the primary tumours across the 4 PDX models was similar however, a few model specific differences were noted (Figure S4). Mesenchymal-associated genes EGFR and RRAS, and an anoikis-associated gene (ILK) had high expression in LuCaP70 and LuCaP105. VIM was uniformly high in all LuCaP105 samples. FN1 had high expression in only LuCaP96, while VCL had high expression in all models except BM18. By contrast, epithelial genes JUP, CDH1, GRHL2, EPCAM, CLDN3, CLDN4, CLDN7 and KLK3 were uniformly highly expressed in all models. Despite being rich in epithelial markers, the tumours appeared to have a hybrid-like phenotype due to upregulation of few mesenchymal genes. In terms of the CSC markers, CD24 was low in LuCaP70, and CD44 was low in BM18. The housekeeper genes were consistently highly expressed in all the models.“
3-In line 165, it is mentioned that 85 samples were analyzed, but I do not understand which samples were these.
An error in the total number of samples used for the pre-screen was observed (mentioned as 85), and has now been corrected. This was an independent error and does not change any of the subsequent analysis and discussion. We have now added a schematic diagram for the experimental workflow (Figure S1), which makes the samples analysed at each stage clearer for the reader.
4-Do the authors have a potential explanation why the number of CTCs was so different among the blood samples?
The large variation observed here in CTC enumeration across blood samples is comparable to that observed with patient samples. A statement to this effect has been added to the discussion section (lines 417-419). We do not currently have any explanation for the variation.
5-The authors should emphasize more in explaining their selection of the 42 gene panel.
This is covered in detail in our previous and referenced in the current manuscript publication (Reference 30, Tachtsidis et al 2019, PMID: 31190270).
Essentially genes were chosen to represent epithelial or mesenchymal phenotypes, in the context of related processes of hypoxia, cancer stem-ness, metabolism, hormone response, and housekeepers. This information is included in the manuscript on lines 193-190; “In this panel, SNAI1, VIM, NOTCH1, EGFR, FN1, SERPINE1, SNAI2, VCL, IGF1R, RRAS, FOSL1, MSN, NRP1, LAMC2, TNC, EMP3 and INHBA are mesenchymal-associated genes and JUP, KRT20, KLK3, CDH1, GRHL2, EPCAM, BMP7, CLDN3, CLDN4, and CLDN7 are epithelial-associated genes [34]. A small number of hypoxia-associated genes (APLN, HIF1A, and BNIP3), cancer stem cell (CSC) markers (CD24 and CD44), hormonal regulation (HR) genes (ESR1, PGR, and TFF1), selected other genes (PPARGC1A, ILK), and HKGs (RPL32, GUSB, TBP, OAZ1 and NONO) were also included in the panel.” It should be pointed out that the requirement for human specificity in primer design meant that some candidate genes in these classes were not included.
6- This finding could be more analyzed: “The remainder of blood samples did not show bias towards an epithelial or mesenchymal state, nor were they similar to their matched primary tumours.”
As the reviewer notes, we have chosen not to interpret “the remainder of the blood samples”. We have reworded this sentence. This group of samples includes those for which there are overall fewer gene expression values, or too few samples with specific gene expression patterns to reliably define a separate phenotype group. While we feel that there is value in including these samples in the published dataset, we wish to avoid overinterpreting this part of the dataset as there it lacks a definitive profile.
Discussion:
- The authors could discuss the limitations and future perspectives of their study in the “Discussion”.
In the discussion, we have discussed the limitations and future perspectives of our study as follows:
“The high RNA expression of both epithelial- and mesenchymal-associated genes in blood samples is consistent with these mixed CTC subpopulations, and could either correspond to hybrid CTCs or a mixture of epithelial and mesenchymal CTCs. …. While analysing pooled expression data, it is important to consider the presence of heterogenous CTC populations, as shown by our CTC staining results.” (lines 461-470)
“We conclude that functional studies of the most strongly and consistently overexpressed genes in CTCs, such as SERPINE1, KRT20, and PPARGC1A, are warranted to allow us to better understand their functional implications in metastasis. This may open a new window for the development of targeted therapies and personalised medicine.” (lines 578-581)
We also discuss the limitation of our study in the conclusion section:
“Caution should be taken while interpreting pooled CTC analysis due to high CTC heterogeneity across the EMP axis.” (lines 700-701)
In response to the reviewer’s comment, we have further added the following to the discussion:
“Interrogation of metastatic deposits using our 42-gene panel could help uncover their EMP status, and allow us to better understand which genes might be involved in establishment of secondary tumours. However, none of these PDX models presented with detectable micro-metastasis in the study time-frame,” (lines 581-584)
Round 2
Reviewer 3 Report
The revised manuscript has significantly been improved. The Reviewer’s comments have been adequately addressed. The current form of the manuscript is now acceptable for publication in “Cancers”.
This manuscript is a resubmission of an earlier submission. The following is a list of the peer review reports and author responses from that submission.
Round 1
Reviewer 1 Report
In this work, the authors investigate EMT-related markers in CTC from mouse PDX prostate cancer models. This is a novel study investigating possible novel markers related to a hybrid epithelial-mesenchymal state in PDX-derived CTCs. The manuscript is interesting and well written, and I consider it can be accepted for publication after addressing some minor comments.
My comments:
Besides tumor weight, is there a correlation between original tumor stage or castration resistance and CTC % positivity? The manuscript would benefit from a discussion of the malignancy status, metastatic capacity of each of the PDX models found.
The heatmap in figure 4 is hard to follow. It is not clear what the legends for each sample mean. Also, the font size for the color legends is very small. Since there are only 4 different PDXs being analyzed, adding another color bar for PDX type would make it easier to draw conclusions.
Page 6, line 161, it says comprising 1 or 2 tumors of each PDX type, when referring to CTCs, the word tumors is confusing.
Line 322, it says “CDH1 in was only observed in 1 of 4 models “, please check.
Few changes in gene expression between CTCs and the tumor were observed in the LuCaP96 PDX, please discuss. What are the tumor characteristics (proliferation, aggressiveness) of this model?
Can the authors suggest a best gene to look at in prostate cancer CTCs? A gene panel? How do these genes compare to other prostate CTC markers in the literature?
In the introduction, the authors mention that CTCs are currently isolated based on the expression of epithelial cell markers EpCAM and epithelial cytokeratins. In the present study, EpCAM levels were high between tumor and CTC samples. Does this data support the use of current CTC analyses based on epithelial marker expression? Please comment.
Author Response
REVIEWER 1
- Besides tumor weight, is there a correlation between original tumor stage or castration resistance and CTC % positivity? The manuscript would benefit from a discussion of the malignancy status, metastatic capacity of each of the PDX models found.
Discussion of the results in the context of source of PDX of each tumour model has been added, as this is potentially relevant to CTC phenotype. There is no obvious relationship with original tumour stage (CTCs from PDX derived from metastatic sites are the lowest and highest CTC % positivity) or castration resistance. [lines 295-302]
- The heatmap in figure 4 is hard to follow. It is not clear what the legends for each sample mean. Also, the font size for the color legends is very small. Since there are only 4 different PDXs being analyzed, adding another color bar for PDX type would make it easier to draw conclusions.
The legend has been amended to improve clarity. The figure legend size has been increased and another color bar added for PDX type as suggested by the reviewer. We have also incorporated a scale to indicate CTC burden. [Figure 4 legend, page 4]
- Page 6, line 161, it says comprising 1 or 2 tumors of each PDX type, when referring to CTCs, the word tumors is confusing.
The sentence has been amended to replace “tumour” with “sample”. [line 166]
- Line 322, it says “CDH1 in was only observed in 1 of 4 models “, please check.
Thanks for spotting this. The first “in” was a grammatical error and has been removed. [line 349]
- Few changes in gene expression between CTCs and the tumor were observed in the LuCaP96 PDX, please discuss. What are the tumor characteristics (proliferation, aggressiveness) of this model?
The LuCaP96 model was developed from tumour at the primary site. By contrast, the other three PDXs are all derived from metastatic sites. This information is now incorporated into the discussion section (also in response to comment 1). [lines 295-302]
- Can the authors suggest a best gene to look at in prostate cancer CTCs? A gene panel? How do these genes compare to other prostate CTC markers in the literature?
Suggested genes would vary depending on the purpose of “looking at prostate cancer CTCs”. A different gene/gene panel would likely be required for enumeration, as compared to an indication of the functional state of the CTCs. Furthermore, our data uses human specific detection of human CTCs in a murine model, which is important to consider when transferring the gene expression profiles to the clinical setting as many will be expressed in non-tumour associated cells as well. We have now described this rationale in the discussion. [lines 277-279]
In the published literature, epithelial-associated genes such as EPCAM and cytokeratins have been used for CTC isolation and detection. Our data supports the use of EPCAM for prostate CTC enumeration (now stated in the discussion). Putative prostate-specific markers that are already being used or proposed to be useful as prostate cancer CTC markers include KLK3, AR, PSMA, PSCA, AMACR, PSAP, TMPRSS2-ERG, PCA3, NKX3.1, HOXB13, KLK4, PSP94, KLK2 and SLC45A3. Of these, only KLK3 was examined in the present study, as the addition of the other 13 genes would have entailed completely rebuilding and validating the gene panel for sensitivity and specificity and this is beyond our resources at the present time. [lines 315-317]
- In the introduction, the authors mention that CTCs are currently isolated based on the expression of epithelial cell markers EpCAM and epithelial cytokeratins. In the present study, EpCAM levels were high between tumor and CTC samples. Does this data support the use of current CTC analyses based on epithelial marker expression? Please comment.
Our data supports the use of epithelial marker expression for CTC analysis in the context of enumeration, as expression of epithelial genes are not lost despite the increase in mesenchymal gene expression. Our study emphasises the hybrid phenotype of CTCs. We have modified the discussion to reflect these conclusions.[lines 315-317]
Reviewer 2 Report
The manuscript entitled “Dysregulated epithelial-mesenchymal plasticity in circulating tumour cells from prostate cancer patient-derived xenograft models” by Sahra Hassan et al. is a research work investigating expressional characteristics of circulating tumor cells in prostate cancer PDX-mouse models. The topic of this work is of interest for basic but also for more translational and clinical research. The amount of animal work is undeniably considerable. Unfortunately, the molecular analyses are insufficient to support the claims and no validation with other analytic methods has been performed. Furthermore, the interpretation of the data is rather superficial and lacks a clear take home message.
Major Comments:
Line 5: problem in the enumeration of the authors. The last author may be missing: “…Erik W. Thompson1,2, *, Elizabeth D. Williams1,3 and *”
The gene annotation nomenclature is not respected throughout the document: please check and correct accordingly (example of guideline: https://academic.oup.com/molehr/pages/Gene_And_Protein_Nomenclature).
The figures provided in this manuscript are mostly composed of one big single panel. The amount of information per figure does not require such a design. I strongly recommend combining the information in denser figures. (As an example, Figures 1, 2 and 3 are phenotypic analyses and could easily be provided as one figure)
Most of the data rely on an array of RT-qPCR analyses (42 genes). Furthermore, the normalization was performed on a single HKG-gene (RPL32), rendering the results very susceptible to variations. Although challenging, a validation of the observation using a different method (e.g. flow cytometry, marker-based magnetic cell separation, western blot, or immunostaining) would have the strength of the reliability of the claims.
Although PDX-models are very relevant for the study of CTCs, it would have been advantageous to check and compare the results with other publically available CTC datasets like high throughput sequencing data (from other models or patients).
The claim of an enriched EMP-signature in CTCs cannot be fully supported by the present data, as the analysis relies on cDNA of a cell mixture (nucleated blood cell fraction) that is very likely to be heterogeneous. Therefore, the EMP signature presented in this study may either result from a transient differentiation state of the CTCs (as stated by the authors) or be the result of a heterogeneous population of mesenchymal and epithelial differentiated CTCs. Closer analyses via FACS, immunofluorescence or in-situ hybridization (RNA-FISH, RNA-Scope, or equivalent) would help to clarify this issue.
Other Comments:
Line 37: Although relevant, the selected reference is quite old and might be replaced or supported by more actual data (e.g. Cancer Today, https://gco.iarc.fr/today/home).
Line 47 to 50: Very long sentence. Might be rephrased to increase comprehensiveness.
Lines 59 to 66: you might decide to edit the flow of ideas to stress a little bit more the fact that CTCs could represent a decisive diagnostic tool to decide and refine treatment strategies. (e.g. PMIDs 31337040, 28819021 or 28191452)
Lines 117 to 120: the data of the control experiments (RPL32 and KLK3 in 4T1 and MDA-MB-231 xenografts) is important information for the reader and should be provided in the supplements.
Lines 147-148: CTC number has already been associated with the size of the tumor of origin (e.g. PMID 0157216 and 30070688). This should be stated correctly.
Lines 149 to 150: The number of CTCs per animal should be provided as a normalized number (e.g. CTCs per ml blood). It is otherwise difficult to compare these results, as the amount of blood analyzed would influence the number of CTCs detected.
Lines 151 to 153: CTC numbers and SD might be provided as a separate graph to increase the visualization of the data (with respective controls if possible).
Figure 3: Regression lines would greatly help the readability of this graph. Please also edit the figure legend to make it more comprehensible. Statistical methods used are not provided.
Lines 161 to 163: it is not clear, where are these tumors located in fig4. If I’m not wrong, there are 7 blood samples clustering here. To avoid misinterpretations, please use a color code here and mention it in the main text.
Lines 164 to 165: same comment
Figure 4:
-There is no scale for the color code “CTCs count”.
-There is no unit for the expression values! Is that Z-score, fold change, …?
- I have my concerns, generating a clustering heat map with samples having as few as 5 measured genes. The comparison with other samples with over 20 genes measured not fair and very likely to generate artifacts. The “mesenchymal” genes are underrepresented in the analyzed cohort. I, therefore, recommend reducing the number of genes used in this clustering analysis, and/or restricting the number of samples to those having enough genes for such a comparison (generally high CTC samples).
- The normalization of a single gene (RPL32) is dangerous here, as it might generate a significant bias. A normalization on several HKG would be therefore recommended here (rendered possible by the reduction of analyzed samples).
Figure 5:
- no statistical test.
- the list of analyzed genes is long. Providing a categorization would greatly help the interpretation of the results.
Lines 200 to 251: this big paragraph is a long enumeration of the differentially expressed genes between Tumor and CTC samples. The aim of this analysis is missing. The redline structuring the end of this manuscript completely lacks. Furthermore, the difference in the “number of detected genes per samples” does not allow any good comparison.
Author Response
REVIEWER 2
Major Comments:
- Line 5: problem in the enumeration of the authors. The last author may be missing: “…Erik W. Thompson1,2, *, Elizabeth D. Williams1,3 and *”
This appears to be a problem with the file generated during the submission process. The authorship list (and associated affiliations) is correct on the submitted manuscript, and there is no ‘and’ present.[line 5]
- The gene annotation nomenclature is not respected throughout the document: please check and correct accordingly (example of guideline: https://academic.oup.com/molehr/pages/Gene_And_Protein_Nomenclature).
We have carefully reviewed the document to ensure gene annotation is correct. Please note that PSA is used when discussing protein expression and KLK3 is used as the gene symbol. [throughout document and supplementary data files]
- The figures provided in this manuscript are mostly composed of one big single panel. The amount of information per figure does not require such a design. I strongly recommend combining the information in denser figures. (As an example, Figures 1, 2 and 3 are phenotypic analyses and could easily be provided as one figure)
We have kept these as single figures as originally submitted given the sample cohorts are different (although related) in each figure, and we consider that combining them is likely to make this confusing. However, if the editor wishes we can combine these figures.
- Most of the data rely on an array of RT-qPCR analyses (42 genes). Furthermore, the normalization was performed on a single HKG-gene (RPL32), rendering the results very susceptible to variations. Although challenging, a validation of the observation using a different method (e.g. flow cytometry, marker-based magnetic cell separation, western blot, or immunostaining) would have the strength of the reliability of the claims.
RPL32 was used to normalise the data as it was the most frequently detected of the predefined 4 house keeping genes measured. The expression of other housekeeper genes varied quite significantly between samples so might add a bias in our results. We can see from our analysis in figure 5 that the other housekeeping gene sometimes do vary significantly between tumour and blood samples. We have also have assessed RPL32 both in the Klijn 675 cell line data set (Nat Biotechnol 2015 PMID: 25485619) and the Nguyen 2018 Nature Communications single cell RNA-seq (PMID: 29795293) dataset, and found it to be one of the most effective normalising genes, based on either minimum variation across all samples (675 cell lines) or minimizing the standard deviation of other genes across all individual cells. Similarly, Chen and colleagues (Prostate 2013, PMID: 23280481) assessed possible housekeeping genes ACTB, GAPDH, and UBB and found that expression levels of ACTB and GAPDH were less stable and weaker among different CTCs compared to UBB in metastatic prostate cancer patients and this used a single gene (UBB) for normalisation.
We agree that orthogonal validation would be ideal, however due to resource issues and the COVID-19 situation preventing further laboratory work, we are not able to perform any additional analyses.
- Although PDX-models are very relevant for the study of CTCs, it would have been advantageous to check and compare the results with other publically available CTC datasets like high throughput sequencing data (from other models or patients).
We have added a section putting our results in the context of clinical studies using single cell RNA-sequencing of CTCs derived from people with prostate cancer to the discussion. [lines 376-389]
- The claim of an enriched EMP-signature in CTCs cannot be fully supported by the present data, as the analysis relies on cDNA of a cell mixture (nucleated blood cell fraction) that is very likely to be heterogeneous. Therefore, the EMP signature presented in this study may either result from a transient differentiation state of the CTCs (as stated by the authors) or be the result of a heterogeneous population of mesenchymal and epithelial differentiated CTCs. Closer analyses via FACS, immunofluorescence or in-situ hybridization (RNA-FISH, RNA-Scope, or equivalent) would help to clarify this issue.
The EMP signature described in this manuscript is solely based on gene expression profiles specifically from the tumour cells. We have achieved this by leveraging the power of the PDX model in which tumour cells are human and the rest of the cells, including the nucleated blood fraction are all of mouse origin. We specifically designed human-specific primer to only measure the tumor cell expression profiles, as described in the manuscript. We acknowledge that because this is not a single cell analysis, we can not rule out that a heterogenous population of CTCs exists which could be a mixture of mesenchymally- and epithelially- differentiated CTCs. This does not however alter the overall interpretation of the data, as the presence of more mesenchymal CTCs would represent an EMT. Unfortunately, due to resource issues and the COVID-19 situation preventing further laboratory work, we are not able to perform the additional analyses suggested by the reviewer. We acknowledge that these experiments would provide useful information by defining the gene expression profiles at the level of individual tumour cells. We have acknowledged that single cell sequencing would shed more light on the EMT status of individual cells in the manuscript. [lines 374-376]
Other comments
- Line 37: Although relevant, the selected reference is quite old and might be replaced or supported by more actual data (e.g. Cancer Today, https://gco.iarc.fr/today/home).
We have replaced this reference with the 2018 GLOBOCAN study related to the website suggested by the reviewer. [line 41, Reference 1]
- Line 47 to 50: Very long sentence. Might be rephrased to increase comprehensiveness.
The sentence has been re-worked to improve clarity. [lines 68-72]
- Lines 59 to 66: you might decide to edit the flow of ideas to stress a little bit more the fact that CTCs could represent a decisive diagnostic tool to decide and refine treatment strategies. (e.g. PMIDs 31337040, 28819021 or 28191452)
Thank you for this suggestion. We have incorporated these references and emphasised this point in the preceding paragraph. [lines 69-72]
- Lines 117 to 120: the data of the control experiments (RPL32 and KLK3 in 4T1 and MDA-MB-231 xenografts) is important information for the reader and should be provided in the supplements.
This data has been added in the supplementary data (Supplementary Table S1).
- Lines 147-148: CTC number has already been associated with the size of the tumor of origin (e.g. PMID 0157216 and 30070688). This should be stated correctly.
We agree that this association has already been documented in some other models and clinical settings, however in this paper we have focused on studies in prostate cancer. We have added to the discussion section a description of what is published regarding the relationship between CTC detection and tumor size. [lines 270-273]
- Lines 149 to 150: The number of CTCs per animal should be provided as a normalized number (e.g. CTCs per ml blood). It is otherwise difficult to compare these results, as the amount of blood analysed would influence the number of CTCs detected.
We have provided the number of CTCs per estimated individual mouse whole blood volume rather than as CTCs per ml, thus the data is normalised to mouse weight and already does take into account the amount of blood analysed. We have revised the figure legend (Figure 3) to clarify this. [Figure 3 legend, page 8]
- Lines 151 to 153: CTC numbers and SD might be provided as a separate graph to increase the visualization of the data (with respective controls if possible).
As noted by the reviewers we have indicated CTC numbers as both median and mean/standard deviation in the text of the main document. Given CTC number for individual samples are already visualised in Figures 2 and 4 (and provided as data in Supplementary Table 2), we do not think another figure is warranted here.
- Figure 3: Regression lines would greatly help the readability of this graph. Please also edit the figure legend to make it more comprehensible. Statistical methods used are not provided.
Adding 5 regression lines to Figure 3 results in a very cluttered graph, and thus we have not opted to adopt this suggestion. The legend for Figure 3 has been edited to improved comprehension and the information regarding the statistical test used has been added. [Figure 3 legend, page 8]
- Lines 161 to 163: it is not clear, where are these tumors located in fig4. If I’m not wrong, there are 7 blood samples clustering here. To avoid misinterpretations, please use a color code here and mention it in the main text.
The reviewer has misread the data in the Figure in this case - only 6 samples cluster with the tumors in Figure 4 (and as stated in the text). It appears the reviewer as misread the code, and confused the one tumour sample that has tightly clustered with 5 of these blood samples. These samples are already coded clearly in Figure 4 using aqua for tumour samples and pink for blood samples. We have revised the figure legend to further clarify the sample coding. [Figure 4 legend, page 9]
- Lines 164 to 165: same comment
We have added a box around these samples to distinguish them from the remainder of the blood samples. [Figure 4, page 9]
- Figure 4:
-There is no scale for the color code “CTCs count”.
-There is no unit for the expression values! Is that Z-score, fold change, …?
- I have my concerns, generating a clustering heat map with samples having as few as 5 measured genes. The comparison with other samples with over 20 genes measured not fair and very likely to generate artifacts. The “mesenchymal” genes are underrepresented in the analyzed cohort. I, therefore, recommend reducing the number of genes used in this clustering analysis, and/or restricting the number of samples to those having enough genes for such a comparison (generally high CTC samples).
- The normalization of a single gene (RPL32) is dangerous here, as it might generate a significant bias. A normalization on several HKG would be therefore recommended here (rendered possible by the reduction of analyzed samples).
The scale for the CTC count has been added and the unit for expression values is now mentioned in the Figure legend. [Figure 4 and associated legend, page 9]
The number of detectable genes of least 5 (ie >10% of genes) was predefined prior to performing the clustering analysis (see Fig 2). We appreciate that the variable number missing value for each samples create potential issues in down stream analyses, however we did not have a rational basis to exclude these samples given we did not want to confine our analysis to only the subset of samples with high CTC values.
The normalisation using a single house keeping gene is discussed above (Reviewer 2 point 4)
Figure 5:
- no statistical test.
- the list of analyzed genes is long. Providing a categorization would greatly help the interpretation of the results.
The statistical test performed has been added to the figure legend. The gene categorisation has been mentioned in the figure legend and they have been ordered to keep genes of same characteristics together for clear visualisation. [Figure 5 and associated legend, pages 10-12]
- Lines 200 to 251: this big paragraph is a long enumeration of the differentially expressed genes between Tumor and CTC samples. The aim of this analysis is missing. The redline structuring the end of this manuscript completely lacks. Furthermore, the difference in the “number of detected genes per samples” does not allow any good comparison.
This section is inevitably somewhat long, but we have tried to make it easier to digest by separating better each gene, grouping genes around their theme (e.g. epithelial, mesenchymal, stem-cell, etc), and around their patterns (changed similarly in most models, etc). Expression levels were only compared in mice in which the gene could be detected, because of the limitations of detection threshold; this is how we are able still to have an accurate comparison and reach statistically significant conclusions. We have added Suppl. Fig S1 to show that, with the exception of KLK-3, the L32 RNA level (raw Ct, higher value = lower amount of RNA) was significantly lower in mice in which each gene was not detected, supporting the notion that there were insufficient CTCs for an accurate measurement; these were omitted and not necessarily considered as lower expression. An improved ending to the paragraph has also been drafted to reflect on the aim of the analysis. [lines 217-263]
Reviewer 3 Report
Overall: The study investigated EMP using four PDX prostate cancer models by comparing paired blood and tumor samples by RT-PCR. While the number of samples is small for making conclusions the study provides a discovery strategy to further investigation of CTC biology in human prostate cancer. EMP has been extensively studied and the results are not surprising. This study would benefit from a validation in human samples using existing gene expression datasets in the public domain.
Specific comments:
HFK panel: Recommend the use of a panel of HKGs – instead of a single gene for normalizing the PCR data.
CTC profiling: Without a spiking experiment (or tracking system) it is hard to conclude the expression profiles are from CTCs vs. leukocytes (gene expression changes observed may be caused by induction or reduction of signaling molecules affecting leukocytes). Recommend providing evidence (e.g. labeling tumor cells in xenograft models) of the origin of gene expression changes observed in the PDX models.
Proof of concept: Validate gene expression results in existing human datasets of prostate CTC / blood samples.
Author Response
REVIEWER 3
- HFK panel: Recommend the use of a panel of HKGs – instead of a single gene for normalizing the PCR data.
As mentioned above (Reviewer 2 Point 4), the other housekeeper genes are not always detectable so RPL32 was used.
- CTC profiling: Without a spiking experiment (or tracking system) it is hard to conclude the expression profiles are from CTCs vs. leukocytes (gene expression changes observed may be caused by induction or reduction of signaling molecules affecting leukocytes). Recommend providing evidence (e.g. labeling tumor cells in xenograft models) of the origin of gene expression changes observed in the PDX models.
As mentioned above in response to Reviewer 2 Major Comment 6, leukocytes are murine in the PDX system and thus will either not be detected or very inefficiently detected using the human specific primers used in these experiments. The robustness of this assay was demonstrated in our previous publication using an mCHERRY tagged human breast cancer cell line xenografting approach (PMID: 31190270). We also have particular confidence in the current study with respect to KLK3, as this gene is specifically human and thus cannot be detected in only murine cells. The nested RT-PCR approach we have designed has been highly optimised to detect human rather than murine transcripts at each stage, the primers for the cDNA preparation, outer primers and inner primers have all been designed to be human-specific.
- Proof of concept: Validate gene expression results in existing human datasets of prostate CTC / blood samples.
We have added discussion of our results in the context of 2 single cell RNA-seq prostate cancer CTC publications to the paper to address this point. [lines 376-389]
Round 2
Reviewer 2 Report
The resubmitted version of the manuscript entitled “Dysregulated epithelial-mesenchymal plasticity in circulating tumour cells from prostate cancer patient-derived xenograft models” by Sahra Hassan et al. is a research work investigating expressional characteristics of circulating tumor cells in prostate cancer PDX-mouse models. Because circulating and disseminated tumor cells have been strongly related to the occurrence of metastasis and deadly patient outcome, this topic is undeniably of interest, not only for basic but also for more translational research. As mentioned in the first review, the amount of animal work is remarkable and the use of PDX models is relevant. However, the whole study relies on one single RT-qPCR based assay. Validations and further experiments supporting the claims are so far lacking. The reliability of the study is therefore unfortunately strongly impaired. Furthermore, the authors only partially answered the “Major Comments” provided in the first revision. As the first version of the manuscript led to a rejection, more profound editing would have been expected to make it eligible for a new revision session. Additionally, although the COVID-19 situation undeniably has lowered the capacity of labs to perform experiments, this should not be used as an excuse to avoid further necessary experiments and serious scientific behaviors.
Reviewer 3 Report
The revisions to the article suffice major questions. The clinical significance is still to be proven but the study offers proof of concept of the utility of the murine model for further investigations on human CTC biology .